# Speos: an ensemble graph representation learning framework to predict core gene candidates for complex diseases

Florin Ratajczak[1], Mitchell Joblin[2], Marcel Hildebrandt[3], Martin Ringsquandl[3], Pascal Falter-Braun [1,4] ✉ & Matthias Heinig [5,6,7] ✉

Understanding phenotype-to-genotype relationships is a grand challenge of 21st century biology with translational implications. The recently proposed "omnigenic" model postulates that effects of genetic variation on traits are mediated by *core*-genes and -proteins whose activities mechanistically influence the phenotype, whereas *peripheral* genes encode a regulatory network that indirectly affects phenotypes via core gene products. Here, we develop a positive-unlabeled graph representation-learning ensemble-approach based on a nested cross-validation to predict core-like genes for diverse diseases using Mendelian disorder genes for training. Employing mouse knockout phenotypes for external validations, we demonstrate that core-like genes display several key properties of core genes: Mouse knockouts of genes corresponding to our most confident predictions give rise to relevant mouse phenotypes at rates on par with the Mendelian disorder genes, and all candidates exhibit core gene properties like transcriptional deregulation in disease and loss-of-function intolerance. Moreover, as predicted for core genes, our candidates are enriched for drug targets and druggable proteins. In contrast to Mendelian disorder genes the new core-like genes are enriched for druggable yet untargeted gene products, which are therefore attractive targets for drug development. Interpretation of the underlying deep learning model suggests plausible explanations for our core gene predictions in form of molecular mechanisms and physical interactions. Our results demonstrate the potential of graph representation learning for the interpretation of biological complexity and pave the way for studying core gene properties and future drug development.

Understanding phenotype-to-genotype relationships is one of the most fundamental problems of current biological research with profound translational implications for questions ranging from human healthcare to biotechnological crop improvement. Genome-wide association studies (GWAS) statistically associate phenotypes with specific variants in genomic loci. This approach has been immensely successful and led to the identification of thousands of variants affecting diverse physiological, molecular, and even psychological

[1]Institute of Network Biology (INET), Molecular Targets and Therapeutics Center (MTTC), Helmholtz Munich, Neuherberg, Germany. [2]Amazon, Seattle, Washington, USA. [3]Siemens Technology, Siemens AG, Munich, Germany. [4]Microbe-Host Interactions, Faculty of Biology, Ludwig-Maximilians-Universität München, Planegg-Martinsried, Germany. [5]Institute of Computational Biology (ICB), Helmholtz Munich, Neuherberg, Germany. [6]Department of Computer Science, TUM School of Computation, Information and Technology, Technical University of Munich, Garching, Germany. [7]German Centre for Cardiovascular Research (DZHK), Munich Heart Association, Partner Site Munich, Berlin, Germany. ✉e-mail: pascal.falter-braun@helmholtz-muenchen.de; matthias.heinig@helmholtz-muenchen.de

phenotypes. The problem of identifying likely causal variants within haplotype blocks is increasingly solved by advanced modeling approaches that integrate GWAS and functional genomic data to identify the genetic variants and genes that are likely causal for the observed phenotypic manifestation[1,2]. However, after solving this issue recent analyses still indicate that even simple traits can have thousands of causal variants[3] distributed uniformly across the genome, and many without obvious connection to the known molecular mechanisms regulating the respective trait[4–9]. This insight raises the conceptual question which molecular mechanisms could give rise to such a highly polygenic architecture and the practical question about how to prioritize proteins as interventional and diagnostic targets. The recently proposed omnigenic model postulates that the effects of genetic variation on a trait are mediated by *core* genes, encoding *core* proteins (hereafter used interchangeably depending on context), whose expression, and ultimately function, directly and mechanistically influence the phenotype, whereas *peripheral* genes and proteins constitute a regulatory network that propagates the effects of genetic variants on the phenotype by modulating core gene expression and function[10,11]. The model postulates that the effects of peripheral proteins converge on relatively few core proteins that have a major influence on the trait;[12] consequently many functional mutations in core genes remain at low frequency in the adult population[8], making their detection in GWAS challenging. While the original authors propose *trans*-eQTLs to infer the underlying genetic network for all diseases[12,13], they admit that the required cohort sizes make this approach impractical[11]. Rare variant sequencing, alternatively suggested to associate core genes to diseases[10], similarly requires very large cohorts and has been criticized as a suboptimal strategy[14].

Conceptually, the impact of peripheral genes and proteins is transmitted to core proteins via 'regulatory networks' that encompass all layers of biological regulation[10], and which we more generically refer to as *molecular networks* to include biochemical modes of regulation. Thus, to identify core disease genes, here we propose an advanced machine learning (ML) approach that utilizes physical and regulatory molecular network information to identify core genes using Mendelian disorder genes as a positive training set. Mendelian disorder genes not only "clearly fulfill the core gene definition"[14], but are examples towards the extreme end of the distribution of core genes, as a single Mendelian disorder gene can cause the disease[14]. Moreover, for nearly all modes of biological regulation increasingly complete reference networks are available that describe biochemical interactions and regulatory effects, e.g. protein-DNA contacts and transcriptional regulation[15–17], protein-protein interactions[18,19] and signaling pathways[20], and human metabolism[21]. While similar information is also available from aggregated small-scale studies and predictions[20,22–24], these are affected by a heavy inspection bias of hypothesis driven approaches and therefore not ideal for reliable identification of hitherto unknown core genes[18,25,26] (Supplementary Figure 1, Supplementary Note 1).

With the uptake of graph representation learning in biomedicine[27], novel options exist to process networks alongside the input features in a joint ML model, thus approaching an in silico representation of biological regulation. First implementations based on random-walks[28–37] or graph neural networks (GNN)[38–44] show promise in predicting 'disease genes', but are often disease specific, depend on hard-coded and partially biased input data, and do not further explore the properties of predicted (core) genes (Supplementary Fig. 2). Moreover, in many machine learning applications ensemble approaches outperform individual models[45,46].

Here we present Speos, an extensible and generalizable positive-unlabeled (PU) ML framework that integrates information from biological networks and multiple biological modalities including gene expression and GWAS data to predict candidate core genes for five groups of common complex diseases. In contrast to previous research,

our framework natively integrates pure PU learning with the power of machine learning ensembles to arrive at an unbiased, data driven prediction of candidate genes. Systematic evaluation of the predicted candidate genes using six external datasets demonstrates that these exhibit key core gene characteristics, impact phenotypic manifestations to a similar extent as Mendelian genes and are enriched for potential new drug targets.

As Mendelian genes display all characteristics of 'strong' core genes[14] we use these as positive labels for a positive unlabeled graph representation learning[27] ensemble (Supplementary Fig. 3). To specify the Mendelian genes corresponding to specific complex diseases, we make use of the mapping established by Freund et al.[47]. It uses standardized clinical phenotype terms that characterize the specific symptoms of complex traits to identify sets of Mendelian genes and groups closely related diseases into Mendelian disorders clusters that are mapped to closely related complex diseases. Thus, predicted genes are specific for these groups of highly related diseases, as the models are trained using labels defined by disease specific standardized phenotype terms. Tissue-specific gene expression and gene-level GWAS summary statistics will be used as input features[3,10]. Proving a gene to be a core gene for complex traits either requires unethical human genetic intervention studies or epidemiological human data from extremely large sample sizes[14]. Therefore, we refer to the predicted genes, which exhibit several expected properties of core genes, such as causality in mice, negative selection and differential disease expression in humans, as "core-like". To identify suitable base classifiers, we first conducted a hyperparameter optimization on the full data set assuming negative labels for unlabeled genes.

## Results
### Performance of base classifiers
Although Speos uses an ensemble-approach to achieve a consensus, the performance of the base classifier is expected to be indicative for the performance of the ensemble. We therefore explored the performance of different commonly used base classifiers in our method selection step (Fig. 1a, Supplementary Figs. 3c, 4). Since it is unknown by which regulatory modalities the effects of peripheral genes are transmitted to core proteins and if these differ among diseases, we tested 35 biological networks (Fig. 1b) selected for their unbiased, systematic construction or strict curation approach. The nodes always represent the full set of protein coding human genes while the edges are sourced from the selected network. In case multiple networks are used simultaneously, edges are typed by source network. Among several GNN base classifiers, the widely used GCN[48] layer, which is limited to one network at a time, convolutes the features of each gene with a nonlinear projection of its immediate (1-step) neighborhood in a given network. The TAG[49] layer is similar to GCN but considers higher-order neighborhoods (3-steps) of any node and can block out unhelpful information. RGCN[50], is the relational equivalent of GCN and can consider multiple networks simultaneously. Lastly, FiLM[51] is similar to RGCN, but uses feature-wise linear modulation[52] to exclude and even override unhelpful neighborhood features based on the center node and the connecting edge. Additional GNN layers performed worse during hyperparameter optimization and were not further included (Supplementary Figs. 5–7, Supplementary Note 2). Lastly, we included Node2Vec[53] (N2V), which uses random walks on the network and techniques developed for natural language processing to embed the network topology into vector space in an unsupervised setting. These N2V-generated vectors can then be used as input features by methods that cannot ingest networks directly like multilayer perceptrons (MLP), logistic regression (LR) and random forests (RF).

We compared the ability of these base classifiers to identify Mendelian disorder genes using a 4-fold cross-validation analysis, and quantified performance on the holdout set using area under the receiver operator curve (AUROC) (Fig. 1a). AUROC is suitable for model

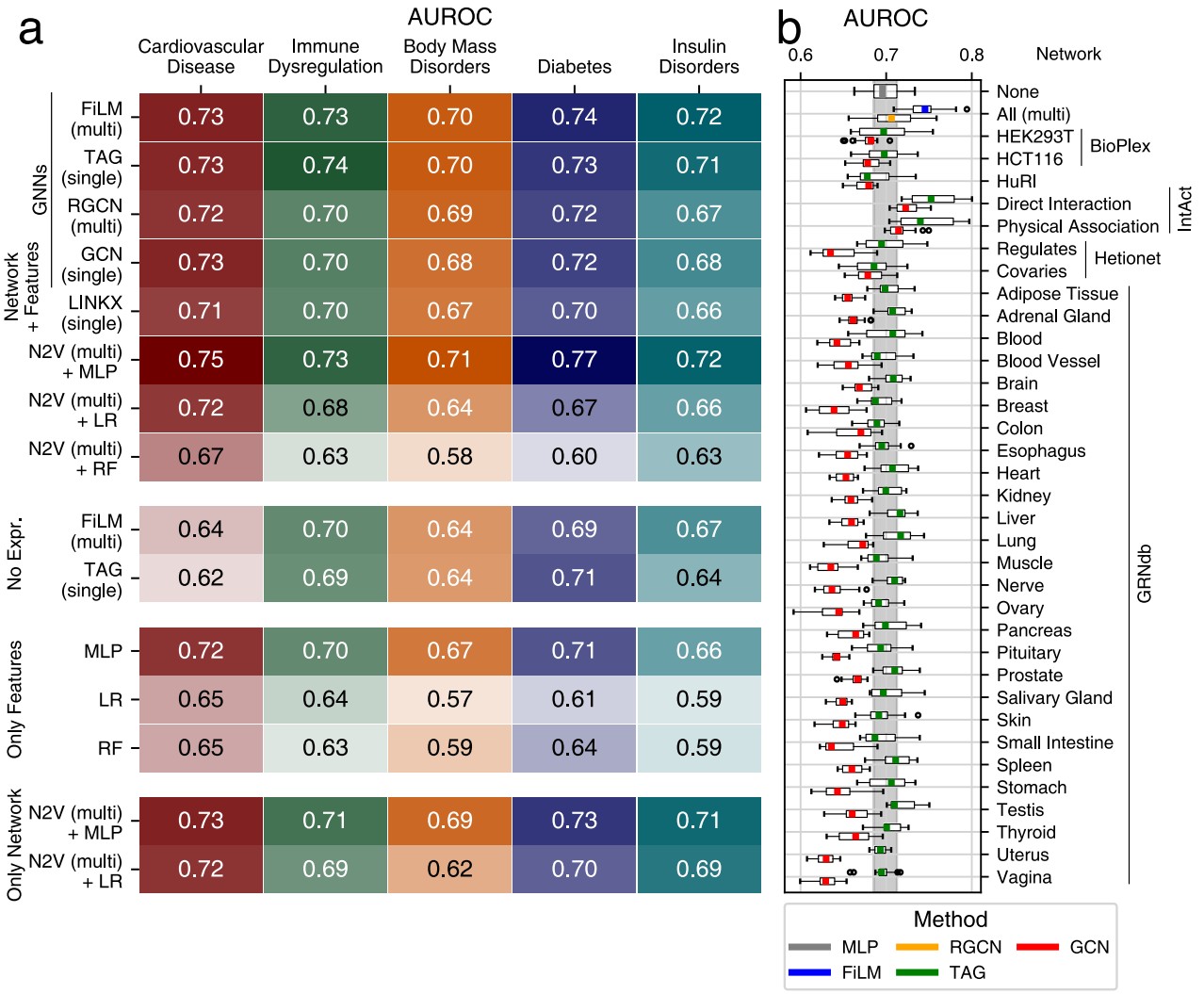

**Fig. 1 | Performance AUROC. a** The mean area under the receiver operator characteristic curve (AUROC) metric (higher is better) over $n = 16$ models for different base classifiers, dataset variants and phenotypes. For AUPRC and mean rank and see Supplementary Fig. 4. Combinations of methods and input data are indicated along the y-axis. The blocks group models using common input data as indicated: Only Network: adjacency matrix/matrices; Only Features: gene expression and GWAS input features but no adjacency matrices; No Expression: GWAS input features and adjacency of individual (single) or multiple (multi) networks; Network + Features: adjacency of individual (single) or multiple (multi) networks, GWAS and gene expression. **b** AUROC of 4 repetitions of a 4-fold cross validation for the indicated individual networks, all networks simultaneously (multi) using the classifier methods indicated by color. The vertical gray area indicates the interquartile range of the MLP, which does not use any network information (uppermost boxplot). Each boxplot is based on $n = 16$ values. Boxes represent the interquartile range, colored bars are medians, whiskers extend at most 1.5 times the interquartile range, and outliers are shown individually.

comparison in PU learning as known positives receive higher predictions than the average unlabeled gene, even though the unlabeled (actual) positives reduce the optimal AUROC score. While many classifiers perform similarly, most methods strongly depend on the input features and the network used. In line with the omnigenic model[10,11], removing tissue-specific gene expression from the input features reduces the performance. The "direct" annotated interaction network from IntAct[22] works well with single-network layers, while the FiLM layer performs well using the union of all networks (Fig. 1b). However, not all networks improve the performance compared to an MLP that does not use any network, likely reflecting the different importance of biological modalities and tissues for different diseases. With GCN, many networks have a detrimental effect on the performance; using TAG, this effect is less pronounced. Equivalently, the FiLM layer improves the performance compared to the RGCN layer when all networks are used simultaneously and tends to predict genes with very high GWAS Z-scores as core genes of cardiovascular disease, consistent with the omnigenic model (Supplementary Fig. 8). However,

overall performance appears to be mostly driven by patterns in gene expression, as ablation experiments suggest (Supplementary Fig. 9). As TAG and FiLM, but not GCN or RGCN, can ignore unhelpful neighborhood information, their increased performance could reflect the fact that not the complete reference network is relevant for disease manifestation and prediction. Intriguingly, in this benchmarking the best performing method (Fig. 1a), N2V + MLP, does not use graph convolutions but embeds all networks simultaneously into vector space using Node2Vec[53] and then feeds these vectors alongside the GWAS and gene expression features into an MLP (Supplementary Note 3).

**Ensemble training and external validation of candidate genes**

The next question in PU learning is how to decide on a suitable threshold for the prediction of a novel previously unknown core-like gene. To address this question, we propose a statistical approach to select thresholds based on nested cross-validation. For this ensemble method, we selected five methods as base classifiers, based on their

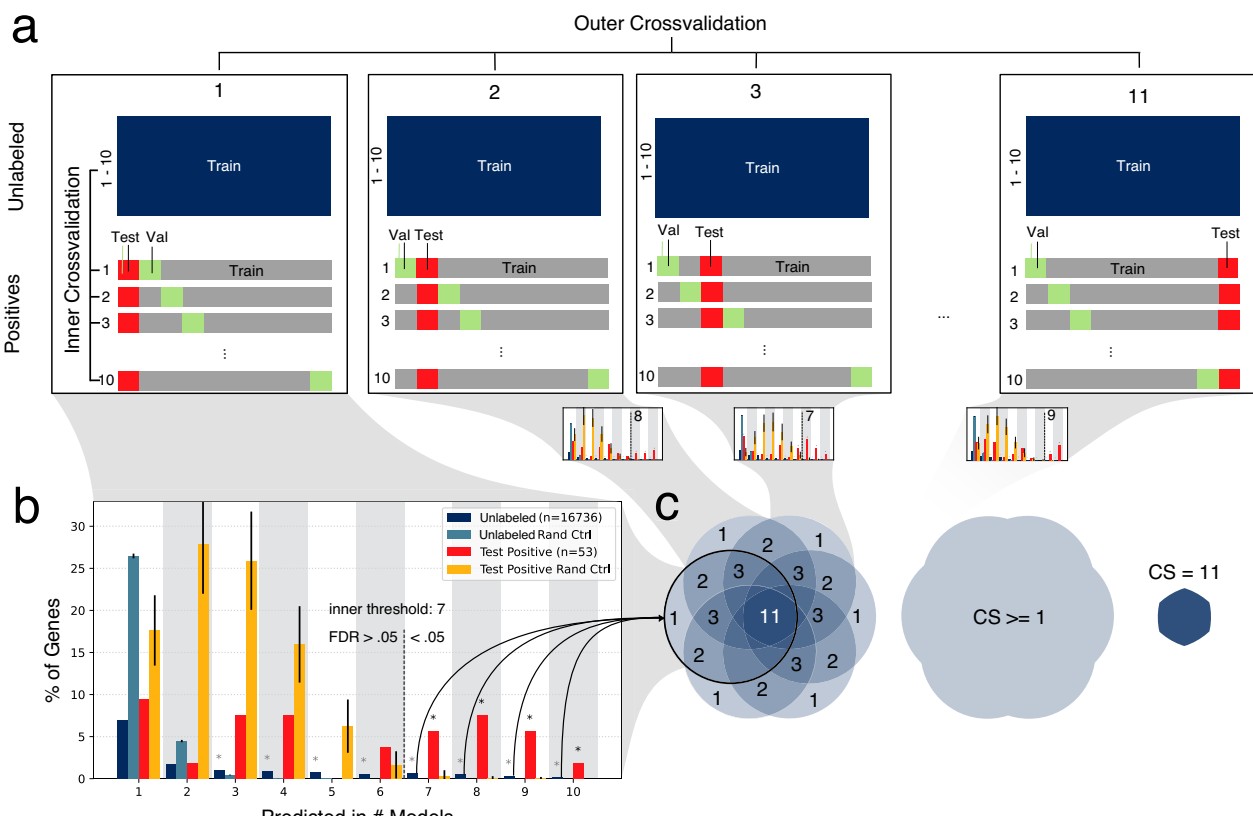

**Fig. 2 | Cross validation ensemble. a** The 11 folds of the outer cross-validation, each with 10 inner cross-validation folds. Each inner cross validation fold corresponds to one ML model. The positives correspond to the Mendelian disorder genes for the given phenotype. Every model within one outer fold has the same positive test set (red square, 9%), but different positive validation sets (green squares, 9%) used for early stopping. All unlabeled genes and 82% of positive genes are used for training for every model of every fold. **b** For each outer fold, the overlap of candidate-predicted unknowns (dark blue bars, $n = 1$ each) and correct predictions of the positive test set (red bars, $n = 1$ each) of the 10 models are compared to random sets of the same size. Mean and standard deviation of the

random sets are shown colored according to the legend (light blue and orange bars, error bars denote one standard deviation based on $n = 1000$ independent drawings). If the observed overlap of correctly classified held out positives is significantly higher than expected by chance (FDR < 0.05, one-sided $t$ tests, Supplementary Data 1, marked with black asterisk), the predicted unlabeled genes of these overlap bins (*inner threshold*) are considered candidate core genes for this outer fold. **c** the candidate genes of each outer fold are aggregated. The Consensus Score (CS) of candidate genes ranges from 1 to 11 and indicates by how many outer folds a given gene is selected as candidate core gene. Genes with CS of 0 are considered non-candidate genes.

performance during method selection: N2V + MLP, which had the best overall performance, FiLM trained on all networks, and TAG trained on IntAct Direct Interaction as best performing GNN-based methods. Despite the lower performance we decided to also include MLP as a baseline classifier that does not use relational network information, and GCN[48], which is regularly used in graph-based problems to ensure comparability with other studies.

We used the selected base classifiers to train the ensembles, which takes the form of a nested cross validation with $m = 11$ outer folds, each comprised of $n = 10$ (inner fold) models (Fig. 2a, Supplementary Fig. 3d). Within each outer fold we statistically assess the agreement of the 10 inner models by selecting an *inner threshold* at which the agreement among the 10 inner models on held out Mendelian genes surpasses random expectation (*FDR* < 0.05; Student's $t$ test, Fig. 2b, Supplementary Data 1). All genes with higher agreement than this *inner threshold* are considered candidate genes of this outer fold. Mendelian genes cannot be predicted as candidates, and hence receive no CS, as they are already known positives and predictions are only computed for unlabeled genes. Since each outer fold predicts one set of candidate genes, the overlap among these sets can be used to assign confidence to each candidate gene using a consensus score (CS) (Fig. 2c), which indicates the number of outer folds predicting a given unlabeled gene to be a candidate. Genes receiving a CS of 0 are non-candidates,

while genes with the highest CS of 11 are the most confident predictions. We aimed to validate the model and our predictions using systematic, orthogonal functional data (Supplementary Fig. 3e).

**Mouse knockout data.** Since core genes are defined as directly contributing to a disease phenotype[10–12], genetic deletion of core genes in mice should cause mouse phenotypes related to the human disease. We therefore investigated if genetic deletion of mouse orthologs of predicted core-like genes across the different CS led to relevant phenotypes more often than expected by chance (Fig. 3a, Supplementary Data 2). Mendelian genes of all five disorders show a significant enrichment, serving as a positive control and benchmark for this validation. From the least conservative (CS ≥ 1) to the most stringent bin (CS = 11) the odds ratio (OR) of mouse knockout genes among the candidate genes increases for all five disease groups. This indicates that, indeed, Speos' CS identifies gene sets of increasing biological relevance and thus can serve as a measure of the quality of predictions. Core-like genes are still significantly enriched when genes with significant GWAS signal are removed (Supplementary Figure 10, Supplementary Data 3), which generally show lower enrichment than the candidate genes (Fig. 3a), indicating that Speos identifies core-like genes outside of significant GWAS genes. Overall, FiLM and TAG predicted gene sets show the highest enrichment and only when all

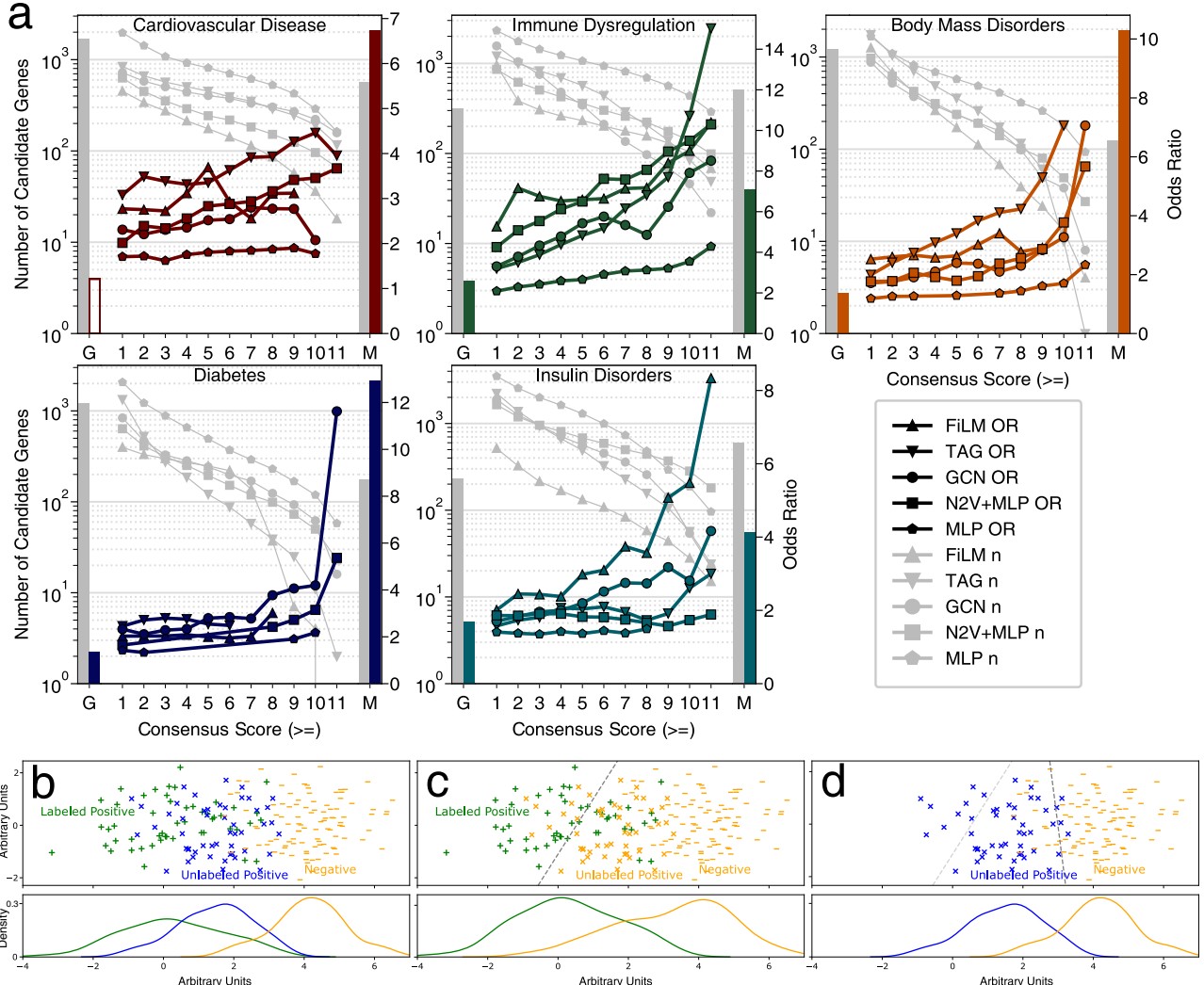

**Fig. 3 | Mouse knockout validation. a** Odds ratio (OR) (right y-axis) for observing disease relevant phenotypes in mice with knock-outs of orthologs of candidate core genes in the indicated consensus score bins (x-axis) of the five classifier methods (colored lines). Gray lines indicate strength of candidate gene sets (left y-axis) in the corresponding bin for the phenotypes as indicated in the panel. Only ORs with an FDR < 0.05 (Fisher's exact test) are shown. Bars to the right (M) and left (G) of each plot indicate set strength (gray) and OR (colored) of Mendelian and GWAS genes for each phenotype. Bars denoting significant ORs are filled, otherwise bars are hollow. Precise *P*-values, FDR, and n for each test are shown in Supplementary Data 2. **b** Illustration of the probabilistic gap according to the "sampled at random with probabilistic gap positive unlabeled" (SAR-PGPU) case from ref. 54.

Labeled and unlabeled positives are drawn from the same underlying distribution, however the label frequency increases towards the more extreme end of the positive distribution, e.g. due to detection bias. We assume this scenario to be true for Mendelian genes as "extreme" core genes[14]. **c** For the internal cross validation on a holdout set (as in Fig. 1a) all unlabeled genes are considered negatives. Consequently, models with the indicated decision boundary (gray dashed line) will perform well. **d** For prediction and subsequent validation of less 'extreme' true, but unknown, core genes indicated by blue labels (Fig. 3a), a model with a decision boundary near the dark gray dashed line is expected to perform well, while the decision boundary from (**b**) (light gray dashed line) is not optimal anymore.

methods show a low performance, as for diabetes, the gap between the methods narrows. For other diseases, represented by cardiovascular disease and body mass disorders, FiLM and/or TAG perform consistently better while GCN, N2V + MLP and MLP remain at the tail end of the distribution. This contrasts with the initial benchmark (Fig. 1a), where N2V + MLP performed best. This discrepancy is likely due to a distribution shift referred to as *probabilistic gap*[54], which here is the consequence of differences between strong Mendelian genes used for training and the additional core-like genes we aim to predict, for which only genetic variants with weaker effects are commonly observed in the population. Because of these differences, (Fig. 3b), the decision boundary that is best suited to recover the 'extreme' Mendelian core genes, i.e., our labeled positives (Fig. 3c), is ill-suited to discover the 'normal' core genes, i.e., unlabeled positives, we aim to discover (Fig. 3d). Importantly, we noticed that the inspection bias of

hypothesis-driven small-scale studies present in the body of literature, and reflected in curated interaction datasets, is amplified in predictions relying on these (Supplementary Fig. 11). Removing the affected networks resolves the bias in the results, yet especially FiLM predictions still validate at similar rates even after removal of the IntAct datasets (Supplementary Fig. 12a, Supplementary Note 4, Supplementary Data 4). Ablation experiments indicate disease-specificity of the mouse knockout gene sets (Supplementary Figure 13, Supplementary Data 5). Furthermore, gene set enrichment analysis for gene ontology (GO) biological processes highlights relevant terms, such as muscle contraction for cardiovascular disease and immune response for immune dysregulation (Supplementary Data 6).

The strong performance in predicting genes with relevant mouse phenotypes clearly demonstrates that Speos identifies disease relevant (core-like) genes. Importantly, at high CS scores, the orthogonal KO

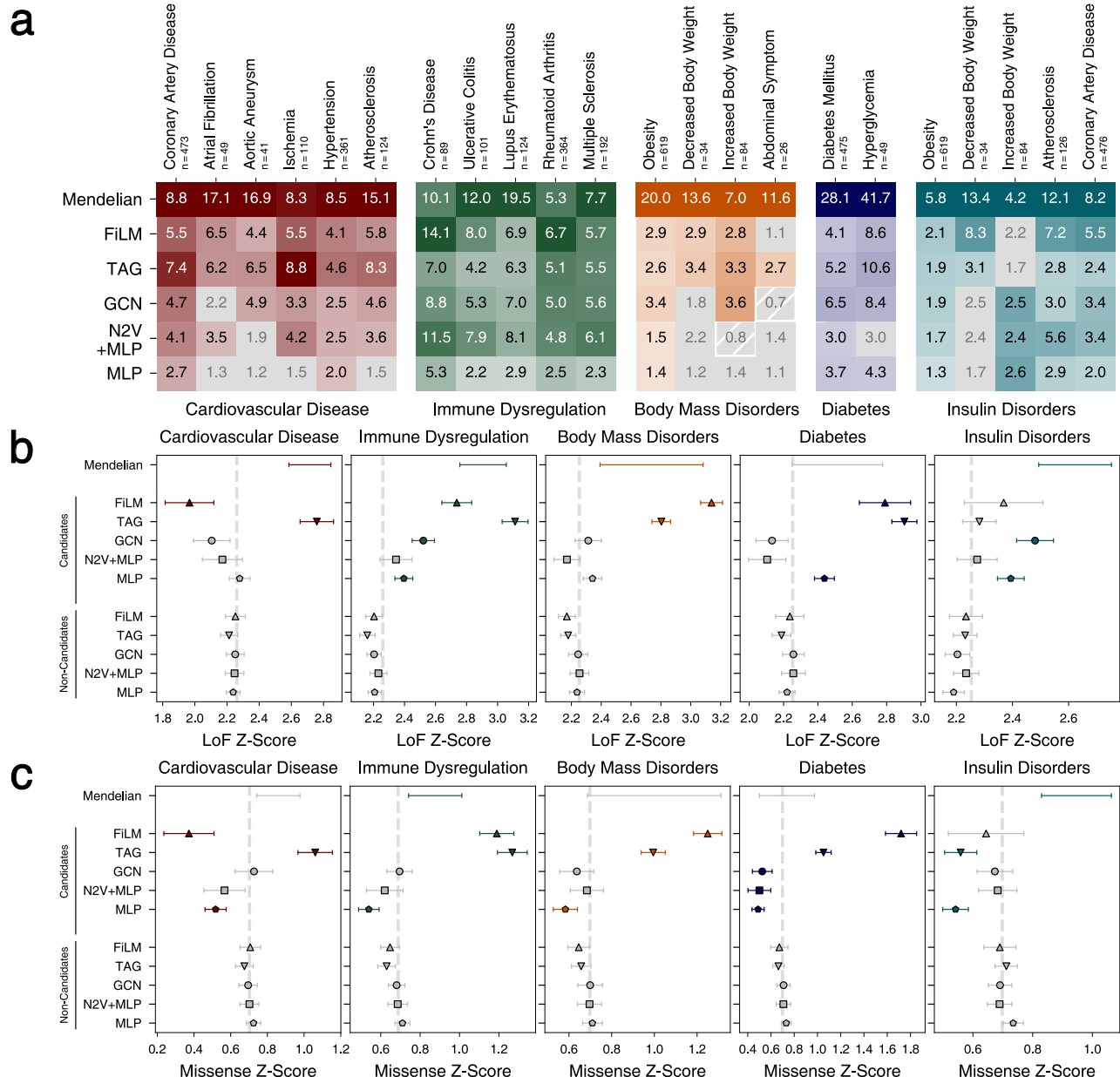

**Fig. 4 | External validation. a** Odds ratios (ORs) of Mendelian genes (first row) and of candidate genes of the five selected methods (rows) for common complex subtypes of the five Mendelian disorder groups. ORs with FDR > 0.05 (Fisher's exact test) in gray. **b**, **c** LoF intolerance and missense mutation intolerance Z-scores of Mendelian genes, and the indicated candidate and non-candidate sets generated by the five methods. Shown are group means and 95% confidence intervals of Tukey's HSD test. Colored symbols and error bars indicate *P* < 0.05 in comparison with respective non-candidate sets; not significant sets in gray. Dashed line indicates the mean across all genes. Precise *P* values, FDR, and n for each test in each panel are shown in Supplementary Data 8, 14, and 15, respectively.

validation rates for sets from all methods except MLP are statistically indistinguishable from the positive control benchmarks for the majority of disease groups (FDR > 0.05, z-test, Supplementary Data 7) and higher than validation rates of GWAS genes (FDR < 0.05, z-test, Supplementary Data 7). Thus, biologically, our predictions are on par with Mendelian genes, which are considered strong core genes. However, even in the lowest bin (CS ≥ 1) genes with disease-relevant mouse phenotypes are enriched for every disorder and every method (FDR < 0.05, Fisher's exact test, Supplementary Data 2), indicating that these sets are meaningful and disease-specific. We therefore include all genes with CS ≥ 1 as candidate genes for the remainder of this work.

**Differential gene expression.** Variation in gene expression can translate into altered enzyme activities and network dynamics and is therefore an important mechanism by which core genes contribute to disease[10,55]. Thus, in disease conditions both Mendelian and predicted core-like genes are expected to be enriched among differentially expressed genes. Indeed, for all disease groups Mendelian genes show a strong enrichment among differentially expressed genes and, again, serve as the reference. The predicted core-like genes are similarly enriched among differentially expressed genes, although the enrichment is weaker for many diseases (Fig. 4a, Supplementary Data 8). This difference is consistent with the notion of 'extreme' and 'normal' core genes and reinforces the idea that both Mendelian genes and core genes underlying the genetic architecture of complex traits can cause phenotypes by loss of function or expression mediated change of activities.

FiLM and TAG predict gene sets with the strongest enrichment in differentially expressed genes with average odds ratios (OR_AV) of 5.4

and 5.0, respectively. Although TAG shows a stronger enrichment of cardiovascular disease subtypes and predicted differential expression-enriched gene sets for 21 out of 22 disease subtypes, FiLM shows the highest $OR_{AV}$ of all methods with especially strong performance in immune dysregulation. The candidate genes produced by GCN show a lower enrichment ($OR_{AV}$ of 4.0), reflecting its initial inclusion as the weakest of the selected GNNs. As before, the performance of N2V + MLP in predicting unknown core-like genes is worse compared to TAG and FiLM. While showing high ORs for some subtypes ($OR_{AV}$ of 3.6), for 5 of 22 disease subtypes the predicted candidate sets show no enrichment for differential expression in disease conditions (FDR > 0.05). The MLP without the Node2Vec node embeddings shows a substantially weaker performance ($OR_{AV} = 2.1$), indicating the importance of network information. Using hypothesis-driven curated interaction datasets differentially impacts the enrichment of predictions for differentially expressed genes for different diseases (Supplementary Fig. 12b, Supplementary Data 9, Supplementary Note 4). Again, GWAS genes among candidates are not the main drivers of enrichment (Supplementary Fig. 14, Supplementary Data 10–13).

Overall, these results indicate that the Mendelian genes tend to be differentially expressed in disease, likely contributing to disease etiology, and that optimized graph convolutional methods such as FiLM and TAG are best suited to generalize this pattern to identify non-Mendelian candidates for core-like genes.

**Loss of function and missense intolerance.** Because core genes directly influence disease phenotypes, these are expected to accumulate protein-function impairing mutations at a lower frequency than regulatory peripheral genes[8,10]. Using ExAc cohort Z-scores[56], we examined this conjecture for two types of functional mutations: loss-of-function (LoF) and missense (Fig. 4b, c, Supplementary Data 14, Supplementary Data 15). Consistent with the omnigenic model, Mendelian genes are enriched for LoF intolerant genes in four out of five disorders. Similarly, candidate core-like genes identified by FiLM and TAG are significantly different from the non-candidates in four out of five disorders. All significant candidate sets are enriched for LoF intolerance, except FiLM predictions for cardiovascular disease genes. For missense mutation intolerant genes we observed overall similar trends (Fig. 4c). Interestingly, the signal from the Mendelian genes is less pronounced and does not reach significance in three of the five diseases. Correspondingly, for four disease groups the signal of the FiLM and TAG predicted core-like genes exceed that of the Mendelian genes and thus presenting the inverse picture than loss-of-function intolerance. Different biological and clinical properties of LoF and non-LoF mutations are well recognized[57] and the observed differences between Mendelian and predicted core-like genes demonstrate that Speos identifies genes with different biological properties than the training set. For cardiovascular diseases, the FiLM predictions again show a significant depletion indicating a potential heterogeneity in the definition of the cardiovascular disease phenotype (Supplementary Note 5, Supplementary Fig. 15, Supplementary Data 16–21).

Taken together, all our analyses strengthen the view of Mendelian genes as 'extreme' core genes, and demonstrate that Speos reliably identifies phenotypically relevant genes with core gene properties.

### Examples and model interpretation

After demonstrating that Speos predicts bona fide core-like genes, we were interested in exploring specific predicted examples to assess plausibility and to understand which aspects of the known biology reflected in the model were most relevant for their prediction as core-like genes (Supplementary Note 6). We selected genes with high CS, which are differentially expressed in at least one disease subtype (Fig. 4a). To explore translational potential, we filtered for genes encoding yet untargeted but druggable[58] proteins and applied model interpretation techniques to investigate gene- and network-level

patterns underlying their prediction as candidates. Both TNFSF15 and IL18RAP are predicted as candidate genes for immune dysregulation by FiLM (CS 11 & 9); the former also by TAG (CS 5) (Fig. 5).

TNFSF15 is differentially expressed in Crohn's disease and ulcerative colitis and its protein product TL1A is part of the tumor necrosis factor superfamily and a ligand for two receptors: DR3 encoded by TNFRSF25, which activates pro-inflammatory signaling, and soluble TR6 encoded by TNFRSF6B, which acts as a non-functional decoy-receptor[59,60]. Increased binding of TL1A to DR3 results in gut inflammation[61,62] and endothelial dysfunction[63,64], while neutralization of TL1A by TR6 down-regulates apoptosis[62]. This ability of TL1A/TNFSF15 to tip the balance of inflammation is mirrored in findings that different genetic variants in- or decrease the risk for Crohn's disease[65–67], ulcerative colitis[66] and inflammatory bowel disease[68]. We investigated influential network-level patterns by assigning importance values to edges using integrated gradients[69]. Model interpretation for TNFSF15 shows that the receptor-ligand relationships with the protein products of TNFRSF25 and TNFRSF6B are among the strongest influences (Fig. 5a) illustrating that Speos' predictions point towards biologically relevant and actionable mechanisms. The model interpretation further suggests that drugs mimicking TR6 can alleviate inflammation by competitively sequestering TL1A and thereby reducing binding of TL1A to DR3. Indeed, monoclonal antibody treatments leveraging this mechanism are in clinical testing and initial results demonstrate a reduction of free TL1A and normalization of pathologically dysregulated gut mucosa[70].

IL−18RAcP encoded by IL18RAP is an accessory protein for the receptor of the proinflammatory interleukin 18 (IL−18) and greatly increases its affinity to its ligand[71]. As such, it can increase the pro-inflammatory effect of IL−18, exacerbating inflammation via the Interferon-γ pathway. IL18RAP is differentially expressed in ulcerative colitis, its expression modulates treatment response in rheumatoid arthritis[72] and it is considered a risk factor for celiac disease[73] and autoimmune thyroid diseases[74]. FiLM's prediction of IL18RAP is highly influenced by its connection to PIGH (Fig. 5b), which is crucial for the first step of the glycosylphosphatidylinositol (GPI) biosynthesis[75]. The GPI glycan supports complex formation between IL−18RAcP and IL-18 receptor which increases proinflammatory signaling[76]. Thus, model interpretation suggests that interfering with the IL-18RAcP−IL-18 receptor interaction reduces dysregulated inflammatory signaling. Indeed, it has recently been demonstrated that cleaving IL−18RAcP using specific antibodies effectively reduces inflammation in human blood cell cultures[77].

Both gene's predictions are strongly influenced by the GWAS input features for the complex forms of the phenotype (Fig. 5c, d). For IL18RAP, high gene expression in whole blood, plasmacytoid dendritic cells (DC) and the spleen are among the strongest influences, which is expected for a factor contributing to autoimmunity[78,79]. This combination of GWAS and disease-specific gene expression are gene-level patterns expected for core genes by the omnigenic model[10–12]. Beyond this, further analyses and examples indicate that Speos finds core-like gene patterns along the entire continuum of evidence combinations, from relying mostly on GWAS features (Fig. 5c), a combination of GWAS and gene expression features (Fig. 5d) to almost exclusively utilizing gene expression features as for obscurin and ITGA7 (Supplementary Figure 16, Supplementary Note 7).

These examples showcase that Speos candidate genes constitute strong core gene hypotheses that are consistent with the omnigenic model. Moreover, model interpretations suggest biochemically and pharmaceutically plausible mechanisms for their impact on disease.

### Speos-candidates are potential drug targets

Since core proteins are defined to directly and causally influence disease phenotypes, countering the respective perturbations with pharmaceutical interventions should improve disease severity and

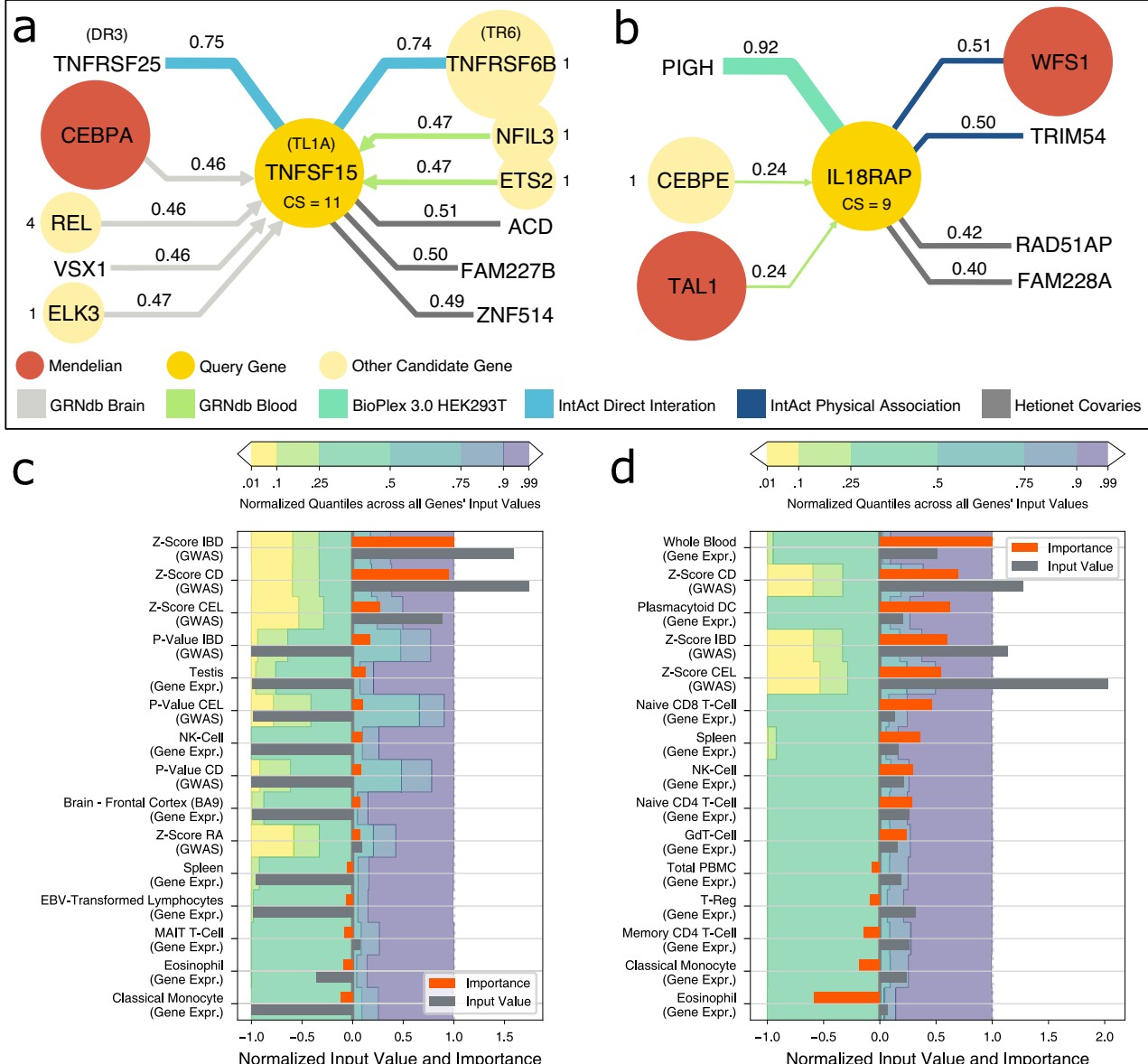

**Fig. 5 | Model interpretation. a** Most important edges for FiLM's prediction of TNFSF15 as candidate gene for immune dysregulation. Shown are HGNC gene symbols, protein symbols are added in parenthesis where necessary. The query gene node is shown in the center, with adjacent relevant nodes in the periphery. Candidate genes are signed with their Consensus Score (CS). The color of the edges denotes the network and the strength of the edge shows the relative importance for the prediction of the query gene which is also written at the edge. Arrowheads indicate direction of edges, undirected edges have no arrowheads. A value of 1 means that it is the most important edge for all models of the ensemble, while a value of 0 indicates that it is the least important edge for every model. Shown are 11 out of 4.3 million edges, 301 of which are in the direct neighborhood of the query gene. **b** Most important edges for FiLM's prediction of IL18RAP as candidate gene for immune dysregulation. Shown are 7 out of 4.3 million edges, 431 of which are in

the direct neighborhood of the query gene. **c, d**: Input feature importance for TNFSF15 and IL18RAP alongside the respective feature's input value, compared to the input values of other genes by the quantile borders in the background. Shown are the 10 features with the strongest positive influence and the 5 features with the strongest negative influence. Negative input values are normalized to the interval [−1; 0] and positive input values to [0; 1] for visualization. Gray bars exceeding the colored areas are either below the 1% quantile or above the 99% quantile of that input feature. Importance values are obtained by integrated gradients and normalized to the interval [−1; 1]. Positive importance values are in favor of the prediction as candidate genes, negative importance values are attributed to features that contradict the prediction. For the input feature importance of surrounding nodes see Supplementary Note 6.

symptoms. To test this prediction systematically, we gathered drug-target-gene interactions from the drug repurposing knowledge graph (DRKG)[80] and assessed the proportion of drug-target encoding genes among the Mendelian and predicted core-like gene sets. Mendelian genes for all five disorders are significantly enriched for genes encoding drug targets (DT), druggable proteins (Dr), and average number of drugs targeting their products (xDC) (Fig. 6, FDR < 0.05, DT and Dr: Fisher's exact test, xDC: U-test, Supplementary Data 22). The enrichments of drug targets (DT) and of the average number of drugs

targeting the encoded proteins (xDC) both suggest that Mendelian genes have been in the focus of drug development. The enrichment of druggable gene products (Dr) among Mendelian genes and predicted candidates could be due to selection biases in the drug discovery process, or may indicate that proteins with binding pockets for substrates or ligands are more likely to be core disease genes that can directly cause disease phenotypes (Fig. 6). Crucially, Speos' predicted core-like genes are similarly enriched in all categories. In contrast to the analyses of biological properties above, the observed enrichments

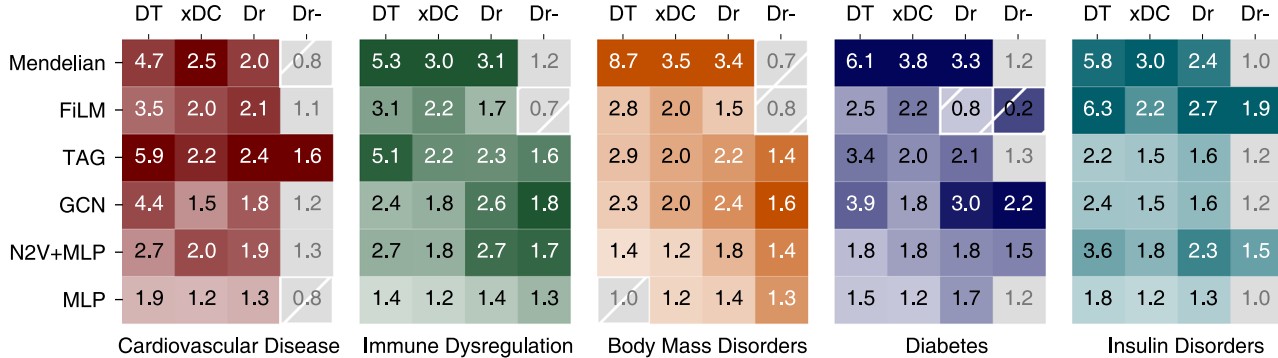

**Fig. 6 | Drug target analysis.** Enrichment of drug targets and druggability in Mendelian disorder genes and indicated candidate gene sets. DT: OR of known drug targets. xDC: Ratio of median number of drug-gene interactions per candidate gene to the median of non-candidates, only genes with drug-gene interactions are considered. Ratios with FDR > 0.05 (U-test) are grayed out. Dr: OR of druggable genes. Dr-: OR of druggable genes, after all drug targets have been removed. Odds Ratios with FDR > 0.05 (Fisher's exact test) are grayed out. For all panels, precise *P* values, FDR, and n for each test are shown in Supplementary Data 22.

are more varied among the methods with each method predicting highly enriched subsets in one or two diseases, except for the network-independent MLP.

In light of the retrospective confirmation of core-like gene products as suitable drug targets, core-like gene-encoded proteins that are not drug-targets yet are attractive candidates for future drug development. However, proteins encoded by Mendelian genes are not enriched for druggable proteins once the established drug targets have been removed (Fig. 6, Dr-), which indicates that the innovative potential of Mendelian gene-products as drug targets has been largely exhausted. In contrast, candidate genes produced by TAG and FiLM, as well as N2V + MLP jointly show a significant enrichment for druggable proteins among the non-drug-targets in all five diseases (Fig. 6, FDR < 0.05, Fisher's exact test, Supplementary Data 18) highlighting potential targets for development of therapeutics for these epidemic disease groups. Removing the IntAct networks results in prediction of significantly more not-targeted druggable (Dr-) genes for immune dysregulation (Supplementary Figure 12c, Supplementary Data 23, Supplementary Note 4b).

## Discussion

Speos is a graph-representation learning framework that predicts novel core-like genes with high external validation rates and properties expected for core disease genes. In developing this framework, we show that all investigated modalities of molecular networks carry relevant information to identify core genes (Fig. 5a, b). At the same time, despite the strong GNN performance in the external biological validation (Fig. 3), we were surprised by the moderate gain from including network information in the initial prediction of held out Mendelian genes (Fig. 1a). This is mirrored by the finding that a substantial part of the information that Speos extracts from molecular networks is encoded in the topology and less so in features of neighboring genes (Supplementary Note 3). This is unexpected as both the omnigenic model as well as the underlying biological thinking predict that the regulatory and biochemical network surrounding a node modulates and impacts its function and activation. The fact that the extensive network information we use does not result in an even greater gain in performance may have a variety of possible reasons that could point to future improvements. Obvious shortfalls are imperfect SNP to gene mappings, residual false-positives and the incompleteness of all network maps[18,81,82]. Similarly, models built on any single network are limited by only accessing a small part and single modality of regulatory links explaining their weak performance. Noteworthy, however, is the observation that learning methods that can selectively ignore link information perform better than those that always consider the complete network neighborhood. We also noticed that the average

shortest path between all genes in the union of all networks is close to 2 and many nodes have degrees exceeding 300 (Supplementary Figure 7c) indicating a very high network density. Likely, in any specific (patho-) physiological setting only a few of these interactions are responsible for dysregulated core protein activity, whereas others matter in other conditions, other tissues, or for processes that do not influence disease etiology. We therefore think that, in addition to a lack of relevant interactions, especially the abundance of disease-context-irrelevant interactions constitute a challenge for learning algorithms and, in fact, for our understanding of network function. For future implementations it may be helpful to include directionality of signaling links for example based on systematic perturbation screens[83–86] and include tissue specificity of edges as explicit features. Therefore, even in the absence of new systematic experimental data, future iterations of this type of work are expected to jointly learn the network and gene representations, thereby improving our understanding of network functioning.

In summary, we show that Speos is able to produce candidate core-like gene sets for different common and complex diseases using Mendelian disorder genes as training examples (Supplementary Data 24). We used a systematic mapping of complex traits to Mendelian genes[47] as input to demonstrate the general power of the method on several diseases. More fine-grained analyses are supported by the framework and can easily be implemented for specific traits of interest by specifying the Mendelian gene sets. This will also allow for determining whether the predicted genes are only relevant for individual traits or show substantial pleiotropy within disease groups. By building on properties predicted by the omnigenic model, we have further shown that these candidate genes are enriched for mouse KO genes, differentially expressed genes, genes intolerant of functional mutations and drug targets, all characteristics that are expected of core genes. The validation results are robust, even when known GWAS genes are removed from the candidates, highlighting the added value of integrating multiple data modalities (Supplementary Figs. 10 and 14, Supplementary Data 3, Supplementary Data 10–13). Furthermore, we show examples of candidate genes that have already been selected for drug development and demonstrate that the model relies on similar data as domain experts. As such, Speos is the first attempt at translating the omnigenic model into an ML framework for predicting and prioritizing core-like genes across several disease areas. Finally, the core-like gene sets predicted by Speos can be used to prioritize genes for experimental validation to provide more definitive evidence of being core genes. We anticipate that our results open the door for a better understanding of core gene attributes and network functioning, and open possibilities for future drug development.

# Methods

## Speos: an ensemble-based PU learning framework

Speos is a fully equipped Python framework which manages the modeling of networks and input modalities as well as the training, evaluation and validation of several machine learning (ML) methods for the prediction of novel core gene candidates. It is available at https://github.com/fratajcz/speos and allows the configuration of experiments, including those reported in this article, without the necessity to write or read any code, facilitating the uptake of computational methods. For our experiments we used the Python version 3.7.12. Furthermore, it is fully extensible, as input data, networks, label and validation sets as well as ML methods can be chosen and added by the user. The following sections describe the modeling, training and validation approaches of Speos as they are used in the experiments shown in the manuscript.

To fully exploit all available data for training and to avoid overestimating the performance of the model, we first conduct a hyperparameter optimization on the full data set assuming negative labels for unlabeled genes to find promising base classifiers and then proceed with an ensemble approach, which we evaluate on independent data sources.

**Selection of base classifiers by cross validation.** We first optimize hyperparameters of base classifiers to identify the best setting of the model architecture based on the performance on immune dysregulation and cardiovascular disease (see Hyperparameter Optimization and Supplementary Note 2). Next, we apply these optimal hyperparameters to all diseases and select the most promising base classifiers for the ensemble approach. We performed a fourfold cross validation with four repeats per fold, each holding out 25% of positive and unlabeled genes. We assume negative labels for unlabeled genes and compare performance by mean AUROC on the holdout set.

**The ensemble approach.** PU learning describes a subdomain of ML approaches for problems where a small set of data points (in our case genes) is labeled positive and the rest of the data points are unlabeled. An intrinsic challenge for this class of problems is that the number of true positives, i.e. the prior class distribution[87–92], is unknown and most classifiers require labels for training. Motivated by the robustness and the performance of ensemble approaches such as bagging in PU learning[39,87,88], we develop a statistical approach to separate candidate genes from non-candidate genes using an ensemble approach[87,88,92] which eliminates the need to predefine[39] or estimate[89] a prior class distribution or to choose an arbitrary cut-off[40,42] on predicted rank distributions. At the heart of Speos is the cross validation ensemble consisting of $m$ outer folds, each containing $n$ models. It is an approach to maximize the utilization of scarce, strong labels and simultaneously exploit the constraints of the genetic domain while satisfying the assumptions of the positive-unlabeled training regimen. In addition to the two-step approach and the ensemble learning, we introduce the following measures to improve PU learning: we designed a specific loss function that upweights positives and we employed a variant of the stochastic gradient descent algorithm that downsamples negatives inspired by refs. 87,93,94.

**Nested cross-validation.** In each outer fold $j \in \{1,2,...,m\}$ the positive labels are split up into the training set $train_j$ and the hold-out test set $test_j$. All positives in $test_j$ are treated as unknown and consequently labeled as negatives (class $y = 0$) during training. The remaining positives are labeled as class $y = 1$ during the training. There are two options to define the hold out sets: (1) define hold out sets containing positives and negatives (i.e. unlabeled examples) or (2) define the hold out sets to only contain positives. In option (1) the held out negatives do not contribute to the loss function during training, whereas in option (2) all

negatives contribute to the loss function during training. Therefore, in option (2) the additional negatives increase the loss if they are unknown positives while they would not contribute to the loss in option (1). In general, the model will only produce supposed false-positive predictions if alternative parameters increase false negative predictions, i.e. decrease sensitivity. Thus, by design of the loss function, such a change of parameters results in a greater loss than "admitting" the "false-positive" predictions of the unlabeled. However, only in option 2 this trade-off is reflected in the overall loss across all negatives used for training. In option 1, the prediction of the held out negatives would have no implication on model parameters, thus failing to induce a loss-guided trade-off between false positives and false negatives. The penalty of making these errors is stronger in option 2 because it was applied to the positively predicted candidate genes that are selected from the training set. Therefore, this leads to even fewer positive predictions overall, i.e. more stringent predictions and thus a more conservative choice than option 1.

Each model $i \in \{1,2,...,n\}$ of the inner cross validation is trained on the entirety of unlabeled genes treated as negatives ($y = 0$) and the subset $train_j$ of positives (Fig. 3a). The set of positives $train_j$ is used for all models in cross validation run $j$, but each inner model $model_{j,i}$ divides it further into $train_{j,i}$ and $val_{j,i}$. Since our holdout $val_{j,i}$ set contains only positives, we quantify overfitting by measuring precision $pr$ on the training data and recall $rec$ on the holdout set for the performance measure $f_1 = 2(pr \cdot rec)$ We train for a maximum of 1000 epochs and always retain model parameters corresponding to the maximum $f_1$, but we stop training if $f_1$ did not improve during 100 epochs from the maximum. Within each outer fold $j$, each model $i$ produces a prediction $\hat{y}_{i,j}^g = 1$ for every gene $g$ if the model prediction is greater than 0.7 and 0 otherwise. The global holdout set of $test_j$ is not accessible for any model in outer fold $j$. We compute the number of concordant predictions for each gene $c_j^g = \sum_{i=1}^{n}(\hat{y}_{i,j}^g)$ for this given cross validation run $j$. Each gene is considered a candidate gene if $c_j^g \geq c^*$ and forwarded to the outer cross validation. The inner threshold $c^*$ is introduced in the next section.

**Calculation of the inner threshold.** To assess if predictions at any threshold have higher concordance than expected by chance, and hence are potentially meaningful, we set aside a global holdout set $test_j$ for every outer fold $j$ (Fig. 2b). We then quantify the overlap of the held-out positive genes $g$ in $test_j$ with concordant predictions of $c^*$ models as $C_j = |\{g|c_j^g = c^* \wedge g \in test_j\}|$. To obtain a background model for the distribution of model overlaps $c_j^g$ we setup $n$ random classifiers $\hat{i}$ that produce the same number of positive predictions $\hat{y}_{i,j} = 1$ as the original models and analogously count the overlaps of their 'predictions' as $\bar{c}_j^g$. We quantify the overlap of the held-out positive genes $g$ in $test_j$ with concordant predictions of $c^*$ random models as $\bar{C}_j = $. This is repeated B = 1000 times to obtain an empirical background distribution. Finally, we compare $C_j$ against $\bar{C}_j$ using a one sample $t$-test for each value of $c^* \in \{1,2,...,n\}$ and apply FDR across the $n$ tests. We choose the inner threshold as the minimal $c^*$ where $C_j$ is significantly greater than the random expectation $\bar{C}_j$ (FDR < 0.05, Student's $t$-test, Supplementary Data 1) or if $\bar{C}_j$ is smaller than 0.1. All positively predicted unlabeled genes which reach at least the inner threshold are considered candidate genes for this outer fold $j$.

**Ensemble prediction.** The $m$ candidate gene sets, one from every fold of the outer cross validation, are assessed for overlapping genes to arrive at a Consensus Score (CS) for every gene. The CS reflects the number of outer folds, which has predicted the gene as a candidate gene. Thus, the CS ranges from 0 to $m$, with 0 indicating that the gene has never been chosen as a candidate and thus is termed a non-candidate gene. Candidate genes have a CS of 1 to $m$, where 1 indicates the least and $m$ the most stringent cutoff.

## Input data

**Labels.** Freund et al.[47] have recently defined 20 classes of Mendelian disorders which resemble common complex diseases. First, they defined lists of standardized clinical phenotype terms for 62 complex traits and used these lists to query the Online Mendelian Inheritance in Men (OMIM)[95] database to retrieve lists of Mendelian disorder genes for every complex trait. Subsequent hierarchical clustering of the retrieved gene lists reveals 20 disease group clusters among the 62 complex traits. Next, Freund et al. gathered GWAS genes for the same 62 traits and compared the GWAS genes with the 20 Mendelian disease clusters. They oberseved that the (common complex) GWAS genes have a significant overlap with phenotypically related Mendelian disease clusters and used this significant association to map GWAS traits to the Mendelian disease clusters. Effectively this establishes a genetic and a symptom based connection between the Mendelian and the common complex forms of diseases. Importantly, Mendelian genes "clearly fulfill the core gene definition"[14]. Thus, we have chosen the Mendelian gene sets proposed by Freund et al. as reliable "known positives" for each disease group. In total we use 598 Mendelian disorder genes for cardiovascular disease, 550 for immune dysregulation, 128 for body mass disorders, 182 for diabetes, and 623 for insulin disorders.

Disease gene prediction is inherently a positive unlabeled (PU) learning problem. Despite this, it is a common approach to compose a supposedly "reliable" negative training set to transform the problem from a PU learning task into semi-supervised classification[39,40]. Precise negative training sets are inherently difficult, if not impossible, to obtain as this requires a positive demonstration that a given gene has no function in a specific, or even a panel of diverse diseases. In light of the modification of genetic risk by genetic variation and environmental factors it requires immense resources to demonstrate the lack of involvement, which renders this approach essentially impossible, if a statistically meaningful negative training set is required. Alternative approaches make assumptions about the nature of disease genes and then define negatives that contrast these assumptions. In different contexts this has shown to lead to very strong biases[96,97], since even inconspicuous household genes host a higher-than-average rate of disease genes[98]. Moreover, using negatives that are most dissimilar to the positives in the input space encourages ML algorithms to find trivial solutions, artificially inflating performance metrics while leading to suboptimal results. In light of these substantial challenges, we decided to use a PU learning approach for core disease gene identification and rely on an internal threshold and external validation to assess precision of the results.

**Nodes and node features.** In the following we define an input matrix $\mathbf{X}^{(0)} \in \mathbb{R}^{n \times p}$, which is used for all experiments where the number of nodes $n$ and the number of features per node $p$ depend on the disease and data availability. The full list of nodes contains $n_{full} = 19220$ protein coding genes. We use tissue-wise median gene expression and GWAS summary statistics as input features, which have to be available for every gene. For the gene expression we use GTEx v7 data which has been obtained via RNASeq across 44 human tissues encoded as median transcript per million (TPM)[99] across all GTEx samples of one tissue. Additionally, we use normalized gene expression levels for 19 different blood cells and total peripheral mononuclear blood cells (PBMC) from the human protein atlas[100]. For each node (gene) this results in a gene expression feature vector of length 63, which is used throughout all analyses that make use of gene expression information, regardless of the input graph. We gathered genome-wide summary statistics from a collection of GWAS of 114 traits that were assembled by the GTEx consortium[101] and are available on zenodo[102]. We converted our protein names/gene symbols to Entrez gene ids and mapped them to the corresponding annotations on the human genome assembly 38. Next we aggregated the GWAS summary statistics on the gene-level using

MAGMA[103] based on the positional overlap of SNPs and gene annotations with an upstream/downstream window of 10 kb. GWAS traits have been mapped to the relevant Mendelian disorder by ref. 47. (Supplementary Data 25, see section "Labels" above). Finally, we used 8 GWAS traits for cardiovascular disease (BW, CAD, HDL, HR, LDL, RBC, PLT, TRIG), 7 for immune dysregulation (CD, CEL, IBD, MS, RA, SLE, UC), 7 for body mass disorders (BMI, BW, HDL, FAT, T2D, TRIG, WHR), 6 for (monogenic) diabetes (BW, HDL, FAT, T1D, T2D, TRIG) and 4 for insulin disorders (BMI, CAD, FG, WHR). We integrate the different sources (GWAS and gene expression) by concatenating the feature vectors from both sources, i.e. number of SNPs per gene, gene-level p-value, gene-level Z-value for every associated GWAS trait and tissue-specific gene expression, for every gene. Genes for which at least one of the mentioned input features could not be gathered are excluded from the analysis. This leaves $n = 17320$ out of $n_{full} = 19220$ for cardiovascular disease, $n = 17042$ for immune dysregulation, $n = 17398$ for body mass disorder, $n = 17460$ for diabetes and $n = 17401$ for insulin disorders (see Supplementary Data 25).

Finally, all input features were scaled by quantiles using scikit-learn's *(v1.0.2)* RobustScaler[104] to facilitate the processing in neural networks. Unlike gaussian normalization, this method is more robust to outliers and extreme skewness of input features.

It is important to point out that Speos is a fully extensible framework, which allows the user to add more features by adding a minimal description and a preprocessing function as outlined here: https://speos.readthedocs.io/en/latest/extension.html#additonal-node-features.

**Edges and types of networks.** Network maps have been generated for different modalities of biological regulation or tissue-specific manifestations. In total, we use 33 different networks in our model. Protein-protein interaction networks (PPI) have been widely used for the analysis of the genetic background of diseases and can be obtained using various methods[105]. Affinity-purification mass spectrometry-based maps predominantly identify stable complexes and contain a mix of direct and indirect associations[105]. For this, we use the systematically collected BioPlex 3.0 HEK293T and BioPlex 3.0 HCT116[19] (accessed 17.3.22). Additionally, we use the Human Reference Interactome (HuRI) (accessed 17.3.22), which has been obtained using a binary multi-assay mapping pipeline, which identifies predominantly directly contacting proteins[18]. Both BioPlex and HuRI were generated in systematic experimental approaches. Additional PPI network data are derived from the IntAct database[22] (accessed 11.5.22), which is a manually curated and annotated source of protein-protein interactions. For our analysis, we use only human interactions and further filter them into two subsets. The first contains all interactions that have been labeled as "Physical Association" including its subcategories, and includes, analogous to AP-MS-based data, direct and indirect protein associations e.g., in large complexes or mediated by rRNAs in the ribosome. The second category "direct interactions" is a strict subset of IntAct Physical Association and requires unambiguous evidence for direct interactions using biochemically purified proteins. In contrast to the systematically collected HuRI and BioPlex datasets, IntAct contains interactions sourced from hypothesis-driven small-scale studies and thus represents the biases inherent to this research approach[18,25,26].

The next type of network that is usually imposed on genes is gene regulatory networks (GRN). Gene regulatory networks are usually directed. Edges run from a transcriptional regulator (the transcription factor—TF) to its target gene. We use 27 tissue-specific GRNs obtained from GRNdb[106] (accessed 29.3.22). These networks have been inferred using enriched TF motifs and RNA-seq expression data of healthy human tissues from GTEx[100,106]. Finally, we use two types of relations from Hetionet (accessed 18.3.22) to define edges[107,108]. The relation "Gene→regulates→Gene" is a non-tissue-specific GRN that has been established from RNA-seq data by the original authors of Hetionet. The

relation "Gene–covaries–Gene" captures coevolutionary patterns of two genes which has been shown to aid in disease gene prioritization[109]. We do not include the third relation that runs between genes, "Gene–interacts–Gene", since we already include several prime candidates for PPIs.

At this point we would like to emphasize that Speos is a fully extensible framework, which allows users to add more adjacency matrices by adding a minimal description as outlined in the documentation: https://speos.readthedocs.io/en/latest/extension.html#adding-a-network.

## Modeling networks for machine learning

All nodes in the used networks represent genes or their encoded protein products, thus the networks represent homogeneous graphs. For each disease, the set of nodes is independent of the network from which the edges are sourced and represents all protein coding genes for which the necessary data is available (see Nodes and node features). For our machine learning approach we model each network as a directed graph. In case of PPI networks, which are inherently undirected, we introduce two edges between two connected genes $gene_a$ and $gene_b$, each going in a different direction, so that the there exist both edges $gene_a \rightarrow gene_b$ and $gene_b \rightarrow gene_a$. In case of gene regulatory networks which are inherently directed, we only model the edges running from the transcription factor (TF) to the modulated genes $gene_{TF} \rightarrow gene_b$, but not vice versa. In the experiments where multiple networks are used simultaneously, each edge is also given a type $r \epsilon R$, which indicates the network the edge is sourced from. This means that two connected genes $gene_a$ and $gene_b$ can, but don't have to be connected by more than one edge of different edge types $r_1$ and $r_2 : gene_a \rightarrow^{r_1} gene_b$ and $gene_a \rightarrow^{r_2} gene_b$. In this case, the graph neural networks used are aware of edge types and treat the edges $e^1$ and $e^2$ seperately. Isolated nodes, i.e., nodes with degree of zero, are included in the experiments, and are convoluted with themselves during graph neural network processing.

## Model architecture

Our general model architecture for most of our base classifiers consists of three consecutively arranged modules: pre message passing, message passing and post message passing (Supplementary Fig. 5a). The pre and post message passing consist of fully connected linear layers, interspersed with exponential linear unit (ELU) activation functions[110]. Their task is to transform the dimensionality from the input dimension to the desired hidden or output layer's dimension. Additionally, they serve as feature extraction layers, where pre message passing extracts and transforms the features so that the message passing can be most efficient, and the post message passing transforms the result of the message passing into a prediction for every gene. Based on hyperparameter optimization, we have chosen two hidden layers plus the input/output layer for both pre and post message passing with a hidden dimension of 50 (see Supplementary Note 2). The message passing is where the actual graph convolutions take place using graph neural network (GNN) layers. Based on hyperparameter optimization (Supplementary Fig. 6) we have chosen two GNN layers, each followed by ELU nonlinearity and instance normalization layers[111].

**GNN-based methods.** GNNs have recently seen a rapid development since Kipf and Welling have proposed their seminal GCN layer[48]. Since then, numerous adaptations of the GCN layer have been proposed, focusing on different weaknesses of the original formulation. We have explored 11 different types of GNNs implemented in PyTorch Geometric[112] (*v2.0.4*) and assessed their suitability for our task. Speos allows the user to choose any of these convolution layers, as well as the number of hidden layers and hidden dimensions of the network. For a detailed account of the graph convolutions we examined alongside with the resulting change in performance, see

Supplementary Note 2. Here we introduce layers that are used throughout our work.

**Graph convolutional network layer (GCN).** The GCN layer is defined as follows:

$$\mathbf{X}^{(t+1)} = \mathbf{D}^{-1/2}(\mathbf{A}+\mathbf{I})\mathbf{D}^{-1/2}\mathbf{X}^{(t)}\mathbf{W}_t \tag{1}$$

where $t$ corresponds to the $t$-th layer of the network. Usually, self-loops are added by adding the identity matrix $\mathbf{I}$ to the adjacency matrix $\mathbf{A}$ which is then normalized by the node degree matrix $\mathbf{D}$. The resulting normalized adjacency matrix is then multiplied with the node feature matrix $\mathbf{X}^{(t)}$ and a trainable weight matrix $\mathbf{W}_t$. The node-specific update rule following this layer definition, also called message passing, is defined as follows

$$\mathbf{x}_v^{(t+1)} = \mathbf{W}_t \sum_{u \in N(v)} \frac{a_{v,u}}{\sqrt{d_v d_u}} \mathbf{x}_u^{(t)}. \tag{2}$$

where $\mathbf{x}_v^{(t+1)}$ is the latent representation of node v at layer $t+1$, which is composed of a linear combination of the latent representations $\mathbf{x}_u^{(t)}$ of nodes at layer t in the neighborhood of $v$, $N(v)$, weighted by an optional weight $a_{v,u}$ of the edge between $u$ and $v$ and the degree of the nodes, $d_v$ and $d_u$. In our experiments, all edges are weighted identically with $a_{v,u} = 1$.

**Topology adaptive graph convolution (TAG).** TAG[49] has been proposed to address the limitation of GCN layers to the 1-hop neighborhood of each node, which implies that the receptive field of GCNs in the graph is directly dependent on the number of layers. TAG contains a hyperparameter $K$ which manages the depth (number of hops) that each TAG layer can reach within the graph. It achieves this by using powers of the adjacency matrix

$$\mathbf{X}^{(t+1)} = \sum_{k=0}^{K} (\mathbf{D}^{-1/2}(\mathbf{A}+\mathbf{I})\mathbf{D}^{-1/2})^k \mathbf{X}^{(t)}\mathbf{W}_{t,k}. \tag{3}$$

We use two layers of TAG with a $K$ of 3, which means that each node's representation can be influenced by nodes 3 hops away for each TAG layer used. It furthermore employs skip-connections between layers so that unhelpful information can be blocked. These skip connections are encoded in the weight matrix $\mathbf{W}_{t,k}$ for $k=0$, as $(\mathbf{D}^{-1/2}(\mathbf{A}+\mathbf{I})\mathbf{D}^{-1/2})^0 = \mathbf{I}$. Like GCN, TAG is not aware of edge types, so it is only applied on individual networks.

**Relational graph convolution (RGCN).** RGCN[50] extends the idea of GCN to be aware of multiple types $R$ of edges between nodes, denoted as $r \in \{0,1,...,|R|-1\}$. Every layer $t$ therefore learns separate weights $\mathbf{W}_r^{(t)}$ of node $v$'s neighborhood for each type of edge $r$ and then sums these up

$$\mathbf{x}_v^{(t+1)} = \mathbf{W}_{root}^{(t)}\mathbf{x}_v^{(t)} + \left( \sum_{r \in R} \sum_{u \in N_r(v)} \frac{1}{|N_r(v)|} \mathbf{W}_r^{(t)}\mathbf{x}_u^{(t)} \right). \tag{4}$$

It furthermore learns edge-independent weights $W_{root}^{(t)}$ that are multiplied with $v$'s node features and added to the neighborhood representation.

**Feature-wise linear modulation convolution (FiLM).** The FiLM[51] GNN layer has been proposed as a generalization of several relational GCN architectures such as relational graph convolution (RGCN) or relational graph attention (RGAT)[113] and is based on the idea of feature-wise linear modulation which has recently been proposed for visual reasoning[52]. It introduces an offset beta and a linear coefficient gamma

for every feature of an incoming message $\mathbf{x}_u^{(t)}$ based on the edge type $r$ and the receiver node $v$

$$\mathbf{x}_v^{(t+1)} = \sum_{r \in R} \sum_{u \in N_r(v)} \sigma(\boldsymbol{\gamma}_{r,v}^{(t)} \odot \mathbf{W}_r x_u^{(t)} + \boldsymbol{\beta}_{r,v}^{(t)}). \tag{5}$$

Where $\sigma$ is a nonlinearity function (Rectified Linear Unit: ReLU) and $\odot$ is the element-wise or Hadamard product. The coefficients $\boldsymbol{\gamma}_{r,v}^{(t)}$ and offsets $\beta_{r,v}^{(t)}$ applied to every message $\mathbf{x}_u^{(t)}$ from node $u$ in the neighborhood of $v$ for each edge type r, $N_r(v)$, are obtained by training a hypernetwork $g$

$$\boldsymbol{\beta}_{r,v}^{(t)}, \boldsymbol{\gamma}_{r,v}^{(t)} = g\left(\mathbf{x}_v^{(t)}, \mathbf{W}_{g,r}^{(t)}\right). \tag{6}$$

so that both $\mathbf{W}_{g,r}$ and $\mathbf{W}_r$ contain trainable parameters. Hypernetworks are neural networks that learn parameters of other neural networks to increase weight-sharing and reduce model complexity and memory requirements[114,115]. In FiLM, $g$ is implemented as a single linear layer. In other words, FiLM $g$ the message that a node $u$ passes to a node $v$ conditioned on the relation $r$ and the latent representation of the receiving node $v$.

**Node2Vec**. Methods like Node2Vec[53] can bridge the gap between graph-native and non-graph methods by first preprocessing the graph, embedding each node into vector space in an unsupervised setting using random walks. These embeddings can then be used by MLPs or regressions as regular input features. We used the *fastnode2vec*[116] (*v0.0.5*) command line interface of *gensim*'s[117] (*v4.1.2*) implementation of Node2Vec with context 5, 100 dimensions, walk length 100 and 500 training epochs on all networks simultaneously. Because Node2Vec does not use edge types, using all input networks is effectively equivalent to using a single network.

**Non-GNN methods**. LINKX[118] is an MLP-based method that first trains MLPs on the input features and adjacency matrix separately and then a third MLP that joins the information of the previous two. It has been proposed to address the shortcomings of GNNs when the first order neighborhood is heterophilous, i.e. the connected nodes do not tend to have the same label. To do so, it trains multiple MLPs: $MLP_A$ is trained directly on the adjacency matrix, using each row of the matrix as feature vectors for the respective nodes. $MLP_X$ is trained on the feature matrix $\mathbf{X}^{(0)}$. Finally, $MLP_f$ uses the concatenated latent representations produced by $MLP_A$ and $MLP_X$ as input and predicts the class label $\hat{y}$. We implemented LINKX in PyTorch[119] (*v1.8.0*) and found that it is prone to overfitting due to the large weight matrix of the first layer of $MLP_A$. We have therefore placed an L1 regularization term on this matrix which we multiply with a factor $\alpha$ and add it to the task-specific loss. We have searched $\alpha$ in powers of ten from $10^0$ to $10^{-5}$ and found the best performance with $\alpha = 10^{-2}$.

The MLP used as a base-classifier resembles the general model architecture outlined above with the number of message passing layers set to 0, only leaving fully connected layers interspersed with ELU nonlinearity. Logistic regression and random forests are implemented using scikit-learn's[104] (*v1.0.2*) LogisticRegression and RandomForestClassifier classes with balanced class weights and sample weights 2 for positives and 1 for unlabeled genes. As they are not able to directly use graph-structured data, they either only use the feature matrix $\mathbf{X}^{(0)}$ (Only Features) or use a concatenation of $X^{(0)}$ and the latent node features obtained via Node2Vec (Network + Features).

**Hyperparameter optimization (HPO)**. A systematic HPO is crucial for most machine learning purposes. We utilize a fourfold cross validation for HPO and report the performance in recovering held out known positives, considering all unlabeled genes as negatives. We assess the area under the receiver operator characteristic curve (AUROC) as

performance metric since we expect an ideal classifier to rank the known positives higher than the average unlabeled gene. To avoid a bias towards a small holdout set given our already small set of reliable positives, each fold trains on 75% of all genes and assesses holdout performance on the remaining 25%. Using the same data for HPO and the validation of the final ensembles would be considered an information leak, resulting in overestimation of model performance. This is why we evaluate the final performance of the ensembles exclusively on additional independent label sets (external validation) which are not present during the HPO. Therefore, the integrity of the training regimen is not compromised. For HPO, we train four models on each fold and report the mean of all 16 resulting models. We first searched for optimal hidden dimension (data not shown), number of hidden layers and type of single-network GNN layer using a selection of networks (Supplementary Fig. 6). Then we searched for the optimal network using all 35 networks and for the optimal multi-network GNN layer using the union of all networks (Supplementary Fig. 7a). See Supplementary Note 2 for detailed results.

**Loss function**. The loss or risk function $L$ measures the goodness of fit of the model and provides the error term from which the gradient is calculated which directly influences the tuning of model parameters via backpropagation. We use class-label 0 for unlabeled genes and class-label 1 for labeled genes and use binary cross entropy, also called logistic loss, as loss function. To reflect the uncertainty of the true label of class 0 and the strength of evidence for our label class 1, we have implemented two mechanisms for loss tuning which we refer to as *dilution* and *amplification*, inspired by ref. [93,103,120]. *Dilution* is a downsampling process where, for each training epoch, we gather a different random subset $U_{sampled}$ sampled uniformly with replacement from all unlabeled genes $U$ so that $|U_{sampled}| = |P_{train}| \cdot d = u^*$ where $P_{train}$ is the set of all positives in $train_{j,i}$ and $d$ is the dilution hyperparameter. This has the advantage that not every unlabeled gene contributes to the loss term in every epoch, allowing unlabeled genes that resemble positive genes to receive a higher prediction, and balancing the contribution of unlabeled and positives to the loss term, eradicating the influence of class imbalance.

The final loss function is composed as follows:

$$L = \sum_{u=0}^{u^*} \frac{BCE(\mathbf{y}_u, \hat{\mathbf{y}}_u)}{d} + a \cdot \sum_{u=0}^{|P_{train}|} BCE(\mathbf{y}_p, \hat{\mathbf{y}}_p) \tag{7}$$

Where $BCE$ stands for binary cross entropy or logistic loss, $a$ is our amplification hyperparameter, $y_u = 0$ and $y_p = 1$. We use $d = 10$ and $a = 2$ in our experiments. For *amplification*, we sum the individual loss terms of positives used for training and multiply it with the amplification factor $a$. This has the effect that false-negative predictions become $a$ times more costly than false-positive predictions. If there exists an unlabeled gene, which is indistinguishable from a known positive, both *dilution* and *amplification* result in a loss that encourages the model to predict both genes as positive (class 1) rather than both as negative (class 0). Although this might lead individual models to overfit to their positive examples in training, ensembles are expected to thrive under these circumstances[121]. We optimize $L$ via gradient descent using an Adam[122] optimizer with learning rate $10^{-3}$.

**Model interpretation**

As candidate genes are predicted by an ensemble, we provide model interpretations based on the average importance of an edge or input feature across the whole ensemble. A related idea of model interpretation has recently been formulated as *model class reliance*[123]. To assess the reliance of the ensemble on certain edges and node features, we gather the respective edges' and nodes' importance using integrated gradients[69] from every model of the ensemble for a query gene $c$. Broadly speaking, integrated gradients assign importance values

based on the change in gradients when input features are substituted with a contrast, usually a vector containing only zeros. For edge importance, this means that we introduce edge weights of 1 for every edge which are then substituted with edge weights of 0. An edge weight of 1 does not alter the message passing and an edge weight of 0 means removing the edge, while gradients backprogated to the respective edge weights can be used for inspection. As we predict the importance based on the gradients backpropagated from gene $c$, the obtained importance values are valid only for the interpretation of the prediction for gene $c$. Each individual model's absolute integrated gradients are minmax scaled to the interval [0,1] across all nodes and edges in the graph. Minmax scaling has the advantage of a comparable output space, but has the tendency to over-emphasize negligible differences in the input space. To alleviate this problematic tendency, we use the mean value of all models' minmax scaled importance values, assuming that an important edge or input feature will repeatedly be close to 1 and an unimportant edge or input feature close to 0, leaving the intermediate values to edges and features that are of ambivalent importance.

Edge Importance $I_{v,e} \in \mathbb{R}$ of edges $e \in E$ for candidate gene $v$ over all models $i \in \{1,2,...,n \cdot m\}$ from all inner and outer folds against a contrast edge weight of 0:

$$I_{v,e} = \frac{1}{n \cdot m} \sum_i minmax_{\forall e \in E}(|IntegratedGradients_i(\mathbf{e},0)|) \quad (8)$$

Note that minmax operates across the set of all edges $E$ (union of all edges across networks in the case of FiLM). Node input feature Importance $I_{v,n} \in \mathbb{R}^p$ of for input features $f \in \mathbb{R}^p$ nodes $n' \in N$ for candidate gene $v$ over all models $m$ from ensemble $M$ against a contrast vector containing only zeros ($0^{\rightarrow} \in \mathbb{R}^p$):

$$I_{v,n'} = \frac{1}{n \cdot m} \sum_i minmax_{\forall n' \in N}(|IntegratedGradients_i(\mathbf{n}',0^{\rightarrow})|) \quad (9)$$

To get a more detailed interpretation of node $v$'s own input features, we also obtain the importance $I_{v,v}$ without removing the sign of the output of integrated gradients and minmax scale it across its own dimensions:

$$I_{v,v} = \frac{1}{n \cdot m} \sum_i minmax_v(IntegratedGradients_i(\mathbf{v},0^{\rightarrow})) \quad (10)$$

This way, the most important feature across all models will receive an importance score close to 1 or −1, depending on the direction of its influence, and the least important feature will receive an importance score close to 0.

For implementation of GNN interpretations we use the interface of PyTorch Geometric[112] (v2.0.4) with the PyTorch[121]-based model interpretation library Captum[124] (v0.4.1).

### External validation and core gene properties

As outlined above, we use all available positive labels for training due to their scarcity. To avoid an information leak between training and validation, we base the validation of our candidate genes on labels sourced from external datasets which are not present during training and hyperparamter optimization but reflect several characteristics of core genes.

**Mouse KO experiments.** We assume that if a gene plays a pivotal role in a disease, severely disrupting the gene's function will result in a phenotype that resembles the disease. To assess this hypothesis, we gathered the same phenotypical queries that ref. 47. used to obtain the labels for the Mendelian genes (Supplementary Data 26). We then used these queries to retrieve a set of genes that, if deliberately knocked out in

mice, produce phenotypes that match the queries using the Mouse Genome Database (MGD)[125,126] (accessed 17.3.22). We used an empty query to get a background set of all available mouse knockout genes. We then translated the mouse genes to their human orthologs using the official MGD mouse-human homolog table (accessed 28.11.22), entries without a human ortholog were discarded, resulting in 16370 genes. For the assessment of candidates, we removed Mendelian genes from the background sets and those genes that were excluded from the predictions due to missing input features, such that the respective intersections of 14116; 13936; 14586; 14541; 14123 (Supplementary Data 25) formed the background sets for the following analysis (Supplementary Data 2 and 3). Next we tested the Mendelian genes of each disease for an enrichment in mouse KO genes against all non-Mendelian genes in the background set, and the candidate genes against all non-Mendelian non-candidate genes in the background set using Fisher's exact test (Supplementary Data 2 and 3). We further tested if restricting the candidate genes to a higher consensus score increases their enrichment. To do so, we tested each CS bin for enrichment against all protein coding non-Mendelian genes with a lower CS. We adjusted the $P$-values of the multiple Fisher's exact tests by FDR correction.

**Differential gene expression.** We gathered differentially expressed genes for subcategories of cardiovascular disease and immune dysregulation by individually querying the following disease subtype in the GEMMA database:[127] coronary artery disease (DOID_3393), Atrial Fibrillation (HP_0005110), aortic aneurysm (DOID_3627), ischemia (DOID_326), hypertension (DOID_10763), atherosclerosis (DOID_1936), Crohn's disease (DOID_8778), ulcerative colitis (DOID_8577), lupus erythematosus (DOID_8857), rheumatoid arthritis (DOID_7148), multiple sclerosis (DOID_2377), obesity (DOID_9970), Decreased body weight (HP_0004325), Increased body weight (HP_0004324), Abdominal symptom (HP_0011458), diabetes mellitus (DOID_9351), hyperglycemia (DOID_4195). Non-human entries were removed. We applied Fisher's exact tests (Supplementary Data 8 & 9) to look for an enrichment of differentially expressed genes in the respective gene sets.

**Gene set enrichment analysis.** We applied gene set enrichment analysis (GSEA) to our candidate gene sets using all using the respective list of 'considered genes' as background. Gene Ontology (GO) Enrichment Analysis performs GSEA based on the GO ontology *biological process*[128,129] (Supplementary Data 6). We obtained the GO annotations through the tool GeneSCF[130].

**LoF and missense intolerance.** We gathered gene-level LoF and Missense Intolerance Z-scores from the ExAc Cohort[56] where a high value indicates a high intolerance for LoF or missense mutations, respectively. In total we obtained Z-scores for 16834 of our $n_{full}$ of 19220 genes, which correspond to 15709 for cardiovascular disease, 15,450 for immune dysregulation, 15781 for body mass disorders, 15787 for diabetes and 15,784 for insulin disorders. We conducted a Tukey's Honestly Significant Difference test (Supplementary Data 8, 9) between Mendelian disorder genes, candidate genes and non-candidate genes.

**Drug targets and druggability.** We obtained drug-gene interactions from the Drug Repurposing Knowledge Graph[80], which has been gathered from a large compendium of databases relating genes, diseases, drugs and several other biomedical domains. We extracted only edges linking drugs and genes and removed edges that have been automatically mined from preprint servers. We considered as drug targets (DT) genes that have at least one edge to any compound and applied Fisher's exact tests (Supplementary Data 22) to look for enrichment of drug targets in our gene sets. To analyze the drug-targeting degree we counted for all drug targets the number of drug-gene interactions. We then applied pairwise Wilcoxon rank sum tests

between the counts of Mendelian disorder genes, candidates and non-candidates and adjusted the *P*-values using FDR (Supplementary Data 22). We report the fold increase of the median drug-targeting degree compared to non-candidate genes (xDC). Genes encoding druggable proteins were obtained from DGIdb[131]. Enrichment for "druggable genes" (Dr) in any set was assessed using Fisher's exact test. To evaluate not-targeted but druggable genes (Dr-), genes encoding products that are already targeted by a drug from the respective gene sets were removed and the remaining druggable proteins tested for enrichment using a Fisher's exact test.

### Reporting summary

Further information on research design is available in the Nature Portfolio Reporting Summary linked to this article.

## Data availability

All data supporting the findings described in this manuscript are available in the article and its Supplementary Information files. All datasets used in this study are already published and were obtained from public data repositories. Edges and Networks: BioPlex 3.0 edge-lists are available at https://bioplex.hms.harvard.edu/interactions.php. HuRI edgelist is available at http://www.interactome-atlas.org/download. Intact edgelist is available at ftp://ftp.ebi.ac.uk/pub/databases/intact/current/psimitab/intact.txt. GRNdb edgelists are available at http://grndb.com/download/. Hetionet edgelist is available at https://github.com/hetio/hetionet/tree/master/hetnet/tsv. Nodes and Features: Full list of human protein-coding genes is available at https://www.genenames.org/download/statistics-and-files/, accessed 18.3.22. Positive labels are available at https://github.com/bogdanlab/gene_sets/tree/master/mendelian_gene_sets, accessed 17.3.22. GWAS summary statistics are available at https://doi.org/10.5281/zenodo.3629742. Tissue-specific median gene expression values are available at https://storage.googleapis.com/gtex_analysis_v7/rna_seq_data/GTEx_Analysis_2016-01-15_v7_RNASeQCv1.1.8_gene_median_tpm.gct.gz. Median gene expression in blood cells is available at https://v19.proteinatlas.org/download/rna_blood_cell.tsv.zip, accessed 17.3.22. External validation: Mouse knockout genes are available at http://www.informatics.jax.org/allele, accessed 17.3.22. Lists of differentially expressed genes were downloaded from https://gemma.msl.ubc.ca/phenotypes.html, accessed 2.8.22. LoF and Missense Mutation intolerance Z-scores are available at ftp://ftp.broadinstitute.org/pub/ExAC_release/release1/manuscript_data/forweb_cleaned_exac_r03_march16_z_data_pLI.txt.gz. List of drug targets are available at https://dgl-data.s3-us-west-2.amazonaws.com/dataset/DRKG/drkg.tar.gz. Lists of druggable genes are available at https://www.dgidb.org/downloads, accessed 24.3.22. For reproducibility, the data can be jointly obtained via Speos' repository: https://github.com/fratajcz/speos or in its processed form from https://doi.org/10.5281/zenodo.7468127.

## Code availability

Speos is open source, implemented in python and available at https://github.com/fratajcz/speos. Config files to reproduce the benchmarks and experiments are also available in that repository. The code can also be obtained via zenodo[132] at https://doi.org/10.5281/zenodo.8416439.

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

## Acknowledgements
F.R. is supported by the Helmholtz Association under the joint research school "Munich School for Data Science—MUDS".

## Author contributions
Conceptualization: M.He., P.F.-B.; Model development and validation analyses: F.R., M.He., P.F.-B.; Method development: F.R., M.J., M.Hi., M.R., P.F-B; M.He., Implementation: F.R.; paper writing: F.R., M.He., P.F.-B.

## Funding

## Competing interests
The authors declare no competing interests.

## Additional information

**Peer review information** : *Nature Communications* thanks the anonymous reviewer(s) for their contribution to the peer review of this work. A peer review file is available.

