## [Peer Review File · Nature Communications]

Speos: An ensemble graph representation learning framework to predict core gene candidates for complex diseasesREVIEWER COMMENTS

Reviewer #1 (Remarks to the Author):

In this work, the authors proposed a framework to predict core genes for diverse diseases with multiple graph representation learning methods. Authors confirmed that mendelian genes display characteristics of 'strong' core genes when they used them as positive labels for a positive-unlabeled graph representation learning. Especially, validation experiments are thoroughly performed.

However, some issues must be addressed before publication.

[Major comments]

- In Introduction, recent network-based embedding studies are neglected. Please cite recent SOTA network-based phenotype prediction studies (e.g.,

Wang, Y. et al. Self-supervised graph representation learning integrates multiple molecular networks and decodes gene-disease relationships. *Patterns* 4, 100651 (2023)).

Also, the main contributions of your study differ from previous research should be highlighted. If possible, prediction performance also should be compared.

- Please provide an overview of the study. It is difficult to comprehend the overall process of the study. A high-level summary or visualization may be helpful to better understand the study.

- I noticed that the supplementary data section lists the data used (nodes: human protein-coding genes, positive labels: mendelian gene set, edges: BioPlex, HuRI, Intact, GRNdb, Hetionet, ...), but it doesn't explain how the integrated graph was constructed (e.g., how to proceed redundant nodes and edges?). Please describe the details. Also, the authors should provide the validity of the constructed graph.

- In addition, it is unclear how GWAS and gene expression data sets were integrated. The authors should provide more information on this to help readers understand the process.

- The authors constructed multiple tissue-specific graphs with different scaffolds. It is unclear whether node features are also tissue-specific for each tissue-specific graph. If so, more information on how the tissue-specific node features were prepared would be helpful.

- A more comprehensive explanation regarding the model train/val/test process should be provided. How were the positive genes distributed (for each disease) during the nested cross-validation? This can be important for understanding the performance of the model and ensuring that it's properly trained and validated.

- The authors incorporated PU learning in their work, which is particularly effective in scenarios where obtaining negative data is challenging. As semi-supervised learning can also address the issue of rare negative data, please clarify the benefits that can be derived from PU learning.

- There could be some confusion regarding the ensemble process presented in this work. To clarify, the ensemble process is actually a combination of prediction results derived from different cross-validation training data, rather than from different graph representation learning algorithms. Therefore, the title and any related expressions should be revised to reflect that the ensemble is not composed of diverse algorithms but of different data folds.

[Minor comments]

* It seems the term both consensus score (CS) and convergence score are used in the mix. (e.g., line 181, Fig.3 (a) legend).

Reviewer #2 (Remarks to the Author):

Ratajczak et al. presented a method Speos, which predicts genes with “core” gene properties by using an ensemble graph representation learning framework. The method aims to address an important problem in complex trait genetics by integrating GWAS, gene expression, and regulatory network information. I have a few major concerns:

1) The authors used mouse knockout datasets to assess the “odds ratios” of candidate core genes (starting from lines 177 and fig 3). The odds ratio of candidate core genes with different CS were compared to GWAS genes and Mendelian genes. While it is nice to see such comparisons, in which candidate core genes seems to be largely in between GWAS genes and Mendelian gene, it is also

necessary to show that: if one excludes GWAS genes and Mendelian genes from the candidate core genes, what the OR looks like?

This is important to show because the authors pointed out that GWAS input features strongly influence their model and the addition of regulatory network information only moderately improves the model. In addition, the two example candidate core genes TNFSF15 and IL14RAP are all previously implicated in GWAS and TWAS studies to be associated with autoimmune traits. Therefore, I expect most candidate core genes to be simply a subset of top GWAS genes. How much value can the complicated Speos model add on top of the GWAS nearest genes and Mendelian disease genes? The worst case is that the candidate core genes are just some GWAS top genes+some Mendelian genes, and that would explain the pattern we see in fig 3.

Similarly, for the drug target analyses (Fig 6), differential expression analyses, and LOF/missense analyses, what the enrichments are like if the authors remove Mendelian and GWAS genes from the candidate gene sets? The drug target enrichment and differential expression enrichments are expected for GWAS genes, therefore naturally drive the enrichments observed. Without showing what additional core genes in addition to GWAS/TWAS/MAGMA and mendelian genes Speos could identify, the usefulness of method is very limited.

2) Core genes are defined as genes that have direct effect on a disease, and it is only defined in a disease phenotype context. What is the rationale for grouping complex diseases into the five complex traits groups? When grouping complex traits into complex trait groups, I find it hard to define core genes. Are core genes specific to only one of the complex traits of the group defined as a core gene to the group? Clear definition of core genes is needed in the paper.

In addition, I also failed to see the disease specificity of core genes in the mouse knockout analyses. Would the core genes in one disease group have increased OR in the rest of the four disease groups?

The definition of the disease groups is also strange such that some diseases are in two different groups, for example, BMI is in two groups. It seems rather random and no clear explanation were given in the manuscript.

3) Results for cardiovascular diseases seems to be an outlier, especially the LOF and missense analyses in Figure 4b and 4c. Simply stating that “..indicating a potentially interesting, but at this point unexplained phenomenon.” is not enough. Did the model simply fail? Are the well-known cardiovascular disease core genes such as APOE or LDLR identified? What are their CS? I don't have confidence in the results, unless the authors could show some evidence that the results make sense. It might also benefit the paper if the authors could go deeper into the cardiovascular disease results.

4) The main text shows the AUROC curve with reasonable numbers, however the AUPRC in Supplement figure 3 is very low (all < 0.15). Core genes should be a small number in comparison to genome wide genes. Is AUPRC better suited than AUROC in this case? Does the low AUPRC mean the predictive value (probability of true positive given positive test) of the model is very low? If so, it is concerning.

5) Relevant to my previous comment, the number of candidate core genes is too big (Figure 3a). Looks like all trait groups have 1000s of core genes, which is impossible. Combining with the low AUROC, does it mean most of the candidate genes are not true core genes?

Minor comments:

Overall, I think the writing is generally clear. However, the intuition behind the complicated method can be better explained in the main text.

**Review response for
"Speos: An ensemble graph representation learning framework to predict core genes for complex diseases"**

We thank the reviewers and editorial staff for their thorough and fair review. All points raised have been addressed with new experimental data and analyses as appropriate. We hope that the editor and reviewers will agree that these changes have clarified and improved the manuscript. Below we respond to each of the points raised and describe how our revised manuscript was modified during the revision.

Reviewer #1 (Remarks to the Author):

In this work, the authors proposed a framework to predict core genes for diverse diseases with multiple graph representation learning methods. Authors confirmed that mendelian genes display characteristics of 'strong' core genes when they used them as positive labels for a positive-unlabeled graph representation learning. Especially, validation experiments are thoroughly performed.

However, some issues must be addressed before publication.

1.1.a In Introduction, recent network-based embedding studies are neglected. Please cite recent SOTA network-based phenotype prediction studies (e.g., Wang, Y. et al. Self-supervised graph representation learning integrates multiple molecular networks and decodes gene-disease relationships. Patterns 4, 100651 (2023)).

AUTHOR RESPONSE

We thank the reviewer for bringing this recent publication to our attention. The Graphene method by Wang et al. uses a two-step procedure that includes a self-supervised pretraining on graphs to obtain initial node embeddings, which are then used for retraining supervised task-specific models. One of the tasks is disease gene prediction. The disease gene prediction is trained using GWAS genes as labels. In our work we predict core genes that are deemed directly causal for disease. GWAS genes and predicted core genes as well as Mendelian disease genes overlap only partially. We show in our validation with mouse knockout data that GWAS genes are not as often directly causal for disease as predicted core genes and Mendelian genes. Therefore, the tasks of predicting GWAS genes and predicting core genes are not directly comparable. The Graphene method was not yet cited in our manuscript as it was published on the day our preprint was finalized. We have now added it to the citations and added the method to Extended data figure 2 (see response to next comment 1.1b).

1.1b Also, the main contributions of your study differ from previous research should be highlighted. If possible, prediction performance also should be compared.

AUTHOR RESPONSE

We have prepared a table for easier comparison with previous research (**Extended Data Fig. 2**). As evident from the table, all other methods rely on assumptions on the prior class probability, which prevents a faithful comparison with Speos. Furthermore, most other methods also rely on biased networks and/or a negative label set, both of which can severely skew the results (see Methods: Input Data: Labels; **Supplementary Note 4**). Avoiding these biases and assumptions was part of the motivation for developing Speos. As it was not possible to sufficiently disentangle most methods from these inherent biases or from the data they were trained on, we have instead opted for a comparison of a variety of base classifiers within our ensemble framework, some of which resemble those used in previous research, such as GCN¹ in the EMOGI framework² or Node2Vec (N2V)³, which at its heart is a random walk with restart (RWR)-based method. This controlled comparison on the same input data allows us to attribute the differences in performance to the underlying base classifiers. Our results clearly indicate increased performance of ensembles comprised of more recently published base classifiers (i.e., FiLM⁴ and TAG⁵) compared to GCN and N2V.

EXCERPT FROM REVISED MANUSCRIPT

“In contrast with previous research, our framework natively integrates pure PU learning with the power of machine learning ensembles to arrive at an unbiased, data driven prediction of candidate genes.”

1.2. Please provide an overview of the study. It is difficult to comprehend the overall process of the study. A high-level summary or visualization may be helpful to better understand the study.

AUTHOR RESPONSE

We would like to thank the reviewer for this suggestion and have included the new **Supplementary Figure SF1** shown below into the manuscript.

EXCERPT FROM REVISED MANUSCRIPT

Supplementary Figure SF1 | Graphical Overview. **a:** The main objective of the study is to predict novel core genes for common complex diseases using Mendelian disorder genes as ground truth “strong” core genes according to the omnigenic model. **b:** We integrate multi-modal data either as feature vectors or as network data. A feature vector containing GWAS and gene expression values is assigned to every gene, genes with missing data are discarded. Genes are connected by several different biological network maps which can be undirected or directed. Each network can be used individually or in conjunction with other networks. **c:** Hyperparameter search testing for variations of networks, input data and methods to identify the methods and settings best suited to recover held out Mendelian disease genes. **d:** Development of a nested cross-validation ensemble that leverages consensus between models in a two-stage process to predict novel core gene candidates from unlabeled genes. The five best performing classifiers from step **c** are used as base classifiers for the ensemble training. **e:** The core gene candidates identified in step **d** are selected for external validation by testing for several criteria that are expected for core genes, such as disease-specificity in knockout experiments and differential expression in a disease context.

1.3 I noticed that the supplementary data section lists the data used (nodes: human protein-coding genes, positive labels: mendelian gene set, edges: BioPlex, HuRI, Intact, GRNdb, Hetionet, ...), but it doesn't explain how the integrated graph was constructed (e.g., how to proceed redundant nodes and edges?). Please describe the details. Also, the authors should provide the validity of the constructed graph.

AUTHOR RESPONSE

We thank the reviewer for bringing up this point, as the procedure by which multiple graphs are merged is very heterogeneous across the literature. In our case, the set of nodes is always identical for a given disease, no matter which molecular network we use to construct the graph. It is the full set of protein coding genes for which we have gene expression and GWAS information. Therefore, no node redundancies can occur. These nodes are then connected by the edges of the selected networks. If only one network is used, then there can be no edge

redundancies. Nodes that are isolated, i.e. have a degree of zero, are still included in the machine learning step. Since self-loops are introduced during the calculation of the neighborhood in a graph neural network (GNN), the GNN acts like an MLP on those isolated nodes. If multiple networks are used at the same time, then the edges are typed, i.e. two nodes a and b can be connected by edge types e^1 and e^2 sourced from different networks. Therefore, again, no edge redundancy occurs, as the graph neural networks used in this case are edge type aware and treat the edges e^1 and e^2 separately.

EXCERPTS FROM REVISED MANUSCRIPT

We have amended the results section with a brief introduction to our network integration approach (amendments in italic):

“Since it is unknown by which regulatory modalities the effects of peripheral genes are transmitted to core proteins and if these differ among diseases, we tested 35 biological networks (**Fig. 1b**) selected for their unbiased, systematic construction or strict curation approach. *The nodes always represent the full set of protein coding human genes while the edges are sourced from the selected network. In case multiple networks are used simultaneously, edges are typed by source network.*”

This procedure is further described in the Methods section in the paragraph “Modeling Networks for Machine Learning”. We have amended the paragraph so that it addresses the concern raised more clearly. Amended phrases are written in italic:

“All nodes in the used networks represent genes or their encoded protein products, thus the networks represent homogeneous graphs. *For each disease, the set of nodes is independent of the network from which the edges are sourced and represents all protein coding genes for which the necessary data is available (see Nodes and node features).* [...] In the experiments where multiple networks are used simultaneously, each edge is also given a type $r \in R$, which indicates the network the edge is sourced from. Thus two connected genes $gene_a$ and $gene_b$ can, but don't have to be connected by *two edges e^1 and e^2 of different edge types r^1 and r^2 : $gene_a \rightarrow^{r^1} gene_b$ and $gene_a \rightarrow^{r^2} gene_b$.* *In this case, the graph neural networks used are aware of edge types and treat the edges e^1 and e^2 separately. Isolated nodes, i.e., nodes with degree of zero, are included in the experiments, and are convoluted with themselves during graph neural network processing. The random walker used in Node2Vec does not discriminate between edge types, thus individual networks are merged to one untyped, unweighted network during the generation of Node2Vec embeddings.*”

1.4 In addition, it is unclear how GWAS and gene expression data sets were integrated. The authors should provide more information on this to help readers understand the process.

AUTHOR RESPONSE

We integrate the data sources by concatenating the feature vectors obtained from every individual source. If a gene is not included in either of the input feature sources, i.e. we don't have neither GWAS information nor gene expression for a given gene, this gene is excluded. Please note that both information sources are genome-wide assays and rare cases of missing information are mostly due to ambiguities in the gene identifiers used in the original data sets. We describe this process in the Methods section in the paragraph titled "Nodes and node features" and have amended the corresponding section to describe the procedure more clearly. Amended phrases are written in italic:

"We integrate the different sources (GWAS and gene expression) by concatenating the feature vectors from both sources, i.e. number of SNPs per gene, gene-level p-value, gene-level Z-value for every GWAS trait included in the trait group under consideration and tissue-specific gene expression, for every gene. Genes for which at least one of the mentioned input features could not be gathered are excluded from the analysis. This leaves $n = 17,320$ out of $n_{full} = 19,220$ for cardiovascular disease, $n = 17,042$ for immune dysregulation, $n = 17,398$ for body mass disorder, $n = 17,460$ for diabetes and $n = 17,401$ for insulin disorders (see Supplementary Table ST13). Finally, all input features were scaled by quantiles using scikit-learn's (v1.0.2) RobustScaler¹⁰⁴ to facilitate the processing in neural networks. Unlike gaussian normalization, this method is more robust to outliers and extreme skewness of input features."

1.5 The authors constructed multiple tissue-specific graphs with different scaffolds. It is unclear whether node features are also tissue-specific for each tissue-specific graph. If so, more information on how the tissue-specific node features were prepared would be helpful.

AUTHOR RESPONSE

Thank you for bringing up this potential for confusion. While, indeed, several of the tested networks are tissue-specific models of gene regulatory wiring, other molecular interaction networks such as HuRI and BioPlex are tissue-independent models of potential physical interactions. Especially for these, the tissue-specificity emerges from the expression features for the nodes.

The node-wise input features contain tissue-specific gene expression features extracted from GTEx and the Human Blood Atlas. These are always the same, irrespective of the graph used in the graph neural network. Tissue specific gene expression is an important predictor of disease relevance, as shown previously for GWAS results^{6,7}, therefore it is informative to keep this full information, regardless of the graph that is used. Moreover, using all expression features across all analyses with different graphs allows us to most adequately judge the influence of the graph on the performance. In response to this comment, we have amended (in italic) the methods section in the paragraph "Nodes and node features":

EXCERPTS FROM REVISED MANUSCRIPT

“We use tissue-wise median gene expression and GWAS summary statistics as input features, which have to be available for every gene. For the gene expression we use GTEx v7 data which has been obtained by RNASeq across 44 human tissues encoded as median transcript per million (TPM)^{100,101}. *For each node (gene) this results in a gene expression feature vector of length 63, which is used throughout all analyses that make use of gene expression information, regardless of the input graph.*”

1.6 A more comprehensive explanation regarding the model train/val/test process should be provided. How were the positive genes distributed (for each disease) during the nested cross-validation? This can be important for understanding the performance of the model and ensuring that it's properly trained and validated.

AUTHOR RESPONSE

During the nested cross validation, each model is trained on 82% of the positive genes and 100% of the unlabeled genes, while 9% of positives are held out for early stopping (validation) and 9% of the positives are held out to calculate the inner threshold, i.e. the number of concordant model predictions necessary to score an unlabeled gene as “high” and thus add to it to the list of candidate genes. These numbers arise from our choice of 10 inner folds plus a globally held-out 11th test set, resulting in 11 total splits. We thank the reviewer for pointing out that the exact percentages of train/val/test split of the positive genes are missing from the description of Fig. 2, and have amended the description accordingly (in italic):

EXCERPTS FROM REVISED MANUSCRIPT

“Every model within one outer fold has the same positive test set (red square, 9%), but different positive validation sets (green squares, 9%) used for early stopping. All unlabeled genes *and 82% of positive genes* are used for training for every model of every fold.”

1.7 The authors incorporated PU learning in their work, which is particularly effective in scenarios where obtaining negative data is challenging. As semi-supervised learning can also address the issue of rare negative data, please clarify the benefits that can be derived from PU learning.

AUTHOR RESPONSE

We thank the reviewer for bringing up this important distinction between PU learning and semi-supervised learning (SSL). As the reviewer has mentioned, SSL is appropriate in scenarios with rare negative labels, PU learning is appropriate in scenarios where robust negative labels are entirely missing. The latter is the case for disease associated genes, as absence of evidence of disease association for a gene cannot be interpreted as evidence of absence of disease association. As illustration, consider a yet unknown disease-causing gene, it might well be the case, that the genomes of affected families with a rare mutation in that gene have simply not been

studied yet. Previous approaches, such as EMOGI², have approached disease gene prediction as an SSL problem by gathering a set of negative labels. However, most approaches arrive at this set of negative labels by relying on assumptions which we find problematic. For example, an often-used set of negatively labelled genes are constitutively expressed genes, or so-called housekeeping genes. However, recent work has shown that these housekeeping genes also harbor disease-inducing mutations⁸, thus making them unsuitable candidates for a robust negative label set. With the multitude of problems arising from finding robust negative labels, we have decided not to use negative labels at all and thus approach the problem as a PU learning problem. We provide a discussion of this topic in the methods section in the paragraph “Labels”:

“Disease gene prediction is inherently a positive unlabeled (PU) learning problem. Despite this, it is a common approach to compose a supposedly “reliable” negative training set to transform the problem from a PU learning task into semi-supervised classification^{39,40}. Precise negative training sets are inherently difficult, if not impossible, to obtain as this requires a positive demonstration that a given gene has no function in a specific, or even a panel of diverse diseases. In light of the modification of genetic risk by genetic variation and environmental factors it requires immense resources to demonstrate the lack of involvement, which renders this approach essentially impossible, if a statistically meaningful negative training set is required. Alternative approaches make assumptions about the nature of disease genes and then define negatives that contrast these assumptions. In different contexts this has shown to lead to very strong biases^{97,98}, since even inconspicuous housekeeping genes host a higher-than-average rate of disease genes⁹⁹. Moreover, using negatives that are most dissimilar to the positives in the input space encourages ML algorithms to find trivial solutions, artificially inflating performance metrics while leading to suboptimal results. In light of these substantial challenges, we decided to use a PU learning approach for core disease gene identification and rely on an internal threshold and external validation to assess precision of the results.”

1.8 There could be some confusion regarding the ensemble process presented in this work. To clarify, the ensemble process is actually a combination of prediction results derived from different cross-validation training data, rather than from different graph representation learning algorithms. Therefore, the title and any related expressions should be revised to reflect that the ensemble is not composed of diverse algorithms but of different data folds.

AUTHOR RESPONSE

We agree that the term “ensemble” is used with different meanings, which might be a source of confusion, which we would be happy to resolve in dialogue with the reviewer. “Ensemble” is vaguely defined in the machine learning literature and, in our understanding, only indicates that the final predictions are synthesized by aggregating the individual predictions of multiple machine learning models but does not define the nature of the aggregation or the models⁹. In fact, the most widely used ensemble method, Random Forests, also only employs a single algorithm as base classifier and also involves the weighting of models based on different data folds. We therefore think that the term ensemble is still valid to describe our method, but we agree with the reviewer

that a more detailed term is appropriate.

To be more clear and avoid potential confusion we have revised most mentions of our ensemble technique to clearly indicate that the defining factor is a cross-validation. Amendments are written in italic:

EXCERPTS FROM REVISED MANUSCRIPT

Abstract: We have developed a positive-unlabeled graph representation-learning ensemble-approach *based on a nested cross-validation* to predict core genes for diverse diseases using Mendelian disorder genes for training.

Line 171: “To address this question, we propose a statistical approach to select thresholds based on nested cross-validation.”

Line 178: “We used the selected base classifiers to train the ensembles, which takes the form of a nested cross validation with $m = 11$ outer folds, each comprised of $n = 10$ (inner fold) models (Fig. 2a).”

Figure 2 Caption: Cross Validation Ensemble.

1.9 (Minor comment) *It seems the term both consensus score (CS) and convergence score are used in the mix. (e.g., line 181, Fig.3 (a) legend).*

AUTHOR RESPONSE

We thank the reviewer for spotting this mistake, we have amended the manuscript to only contain the term consensus score.

Reviewer #2 (Remarks to the Author):

Ratajczak et al. presented a method Speos, which predicts genes with “core” gene properties by using an ensemble graph representation learning framework. The method aims to address an important problem in complex trait genetics by integrating GWAS, gene expression, and regulatory network information. I have a few major concerns:

2.1a The authors used mouse knockout datasets to assess the “odds ratios” of candidate core genes (starting from lines 177 and fig 3). The odds ratio of candidate core genes with different CS were compared to GWAS genes and Mendelian genes. While it is nice to see such comparisons, in which candidate core genes seems to be largely in between GWAS genes and Mendelian gene, it is also necessary to show that: if one excludes GWAS genes and Mendelian genes from the candidate core genes, what the OR looks like?

AUTHOR RESPONSE

We thank the reviewer for bringing up this important point, which we should have made clearer in

the manuscript. Most importantly, the candidates are exclusively sourced from the unlabeled genes and therefore, by definition, cannot contain any already known Mendelian genes (Supplementary Figure SF1). Thus, the high odds ratios are due to Speos finding novel core genes which resemble Mendelian genes in these validation criteria. GWAS genes on the other hand can be either Mendelian genes, or classified as new core gene candidates, or classified as non-candidates. We have defined GWAS genes, by extracting significant SNPs ($p < 5e-8$) from the GWA studies, from which we derived our input values (see Supplementary Table ST13) and then mapped them to the closest protein coding gene using GRCh38 gene coordinates. Since we are interested in finding novel core genes, which exclude known Mendelian genes, all GWAS genes that are also Mendelian genes are not part of the unlabeled candidates. Thus, these cannot be predicted as candidate core genes, and thus cannot contribute to the OR of the predictions. As Fig. 3 shows, the odds ratio of mouse knockout genes among the GWAS genes is generally low and it is therefore unlikely that only GWAS genes among the predicted core genes are responsible for the much higher odds ratio of the candidates. Although we would consider it still meaningful if Speos only prioritized mechanistically relevant core genes from all GWAS candidates, this is not the case: To further address this, we evaluated if the odds ratios of the candidate genes are due to a high proportion of GWAS genes. In a new analysis we have removed all GWAS genes from the predicted candidate genes (see new **Supplementary Fig. SF3** shown below and new **Supplementary Table ST3**). While the candidates of some methods (e.g. N2V+MLP for Immune Dysregulation) show slightly lower ORs in this setting, the general trend persists that predicted core genes have higher ORs than the GWAS and (generally) lower ORs than the Mendelian genes. Thus, consistent with the omnigenic model, some but not the majority of our candidates are also GWAS hits, and GWAS and non-GWAS candidate-core genes show high external validation rates.

EXCERPTS FROM REVISED MANUSCRIPT

We have amended the results section to reflect these findings:

Results, Mouse Knockout Validation:

“Core gene candidates are still significantly enriched when genes with significant GWAS signal are removed (Supplementary Fig. SF3, Supplementary Table ST3), which generally show lower enrichment than the candidate genes (Fig. 3a), indicating that Speos identifies core genes outside of significant GWAS genes.”

Discussion:

“The validation results are robust, even when known GWAS genes are removed from the candidates, highlighting the added value of integrating multiple data modalities (**Supplementary Figures SF3 & SF5, Supplementary Tables ST3, ST10-ST13**).”

Supplementary Figure SF3 | Mouse Knockout validation with GWAS Genes removed from Candidates. Odds ratio (OR) (right y-axis) for observing disease relevant phenotypes in mice with knockouts of orthologs of candidate core genes in the indicated convergence score bins (x-axis) of the five classifier methods (colored lines). Gray lines indicate strength of candidate gene sets (left y-axis) in the corresponding bin for the phenotypes as indicated in the panel. Only ORs with an FDR < 0.05 (Fisher’s exact test) are shown. Bars to the right (M) and left (G) of each plot indicate set strength (gray) and OR (colored) of Mendelian genes and GWAS genes for each phenotype. Filled bars represent ORs with an FDR < 0.05, otherwise bars are hollow. Precise P-values, FDR, and n for each test are shown in **Supplementary Table ST3**.

Furthermore, we have amended the results section to emphasize that candidate genes are only sourced from the unlabeled genes and cannot contain already known Mendelian genes (changes in italic):

“Within each outer fold we statistically assess the agreement of the predictions for the outer hold-out (test) set and the unlabeled genes of the 10 inner models. Using the outer fold hold-out set, we select an inner threshold at which the agreement among the 10 inner models on held out Mendelian genes surpasses random expectation (FDR < 0.05; Student’s t-test, Fig. 2b, Supplementary Table ST1). All unlabeled genes with higher agreement than this inner threshold are considered candidate genes of this outer fold. Mendelian genes cannot be selected as candidates, as they are already known positives and predictions are only computed for unlabeled genes.”

2.1b This is important to show because the authors pointed out that GWAS input features strongly influence their model and the addition of regulatory network information only moderately improves the model. In addition, the two example candidate core genes TNFSF15 and IL14RAP are all previously implicated in GWAS and TWAS studies to be associated with autoimmune traits. Therefore, I expect most candidate core genes to be simply a subset of top GWAS genes. How much value can the complicated Speos model add on top of the GWAS nearest genes and Mendelian disease genes? The worst case is that the candidate core genes are just some GWAS top genes+some Mendelian genes, and that would explain the pattern we see in fig 3.

AUTHOR RESPONSE

As demonstrated in response to comment 2.1a Speos does not simply predict GWAS genes as core genes but identifies gene sets that are complementary to Mendelian and to GWAS genes. Core gene predictions for each individual candidate gene rely on different features that may vary from case to case. To illustrate this, we have intentionally selected examples that are based on different types of features (GWAS and expression). As for the two highlighted candidate genes, TNFSF15 and IL18RAP (not IL14RAP), the reviewer is correct that the GWAS input features show strong influence on the prediction as candidate gene. However, the fact that predictions are driven by GWAS features is not the norm. As shown in ED Fig. 7, only a small fraction of high predictions in held out genes co-occur with a strong GWAS signal, while most of the genes which receive a high prediction have moderate to average GWAS signal. This is also mirrored in the two showcased genes for cardiovascular disease, OBSCN and ITGA7 (Extended Data Fig. 10, Supplementary Note 7), which are mostly driven by their gene expression input features, with low influence of their GWAS features. We therefore think that the integrative scoring of Speos adds substantial value by using GWAS signal without overly relying on it.

We have therefore amended the results section of the manuscript (changes in italic)

EXCERPTS FROM REVISED MANUSCRIPT

*“[...] Equivalently, the FiLM layer improves the performance compared to the RGCN layer when all networks are used simultaneously and tends to predict genes with very high GWAS Z-scores as core genes of cardiovascular disease, consistent with the omnigenic model (Extended Data Fig. 7). However, overall performance appears to be mostly driven by patterns in gene expression, as ablation experiments suggest (**Supplementary Fig. SF2**).”*

2.1c Similarly, for the drug target analyses (Fig 6), differential expression analyses, and LOF/missense analyses, what the enrichments are like if the authors remove Mendelian and GWAS genes from the candidate gene sets? The drug target enrichment and differential expression enrichments are expected for GWAS genes, therefore naturally drive the enrichments

observed. Without showing what additional core genes in addition to GWAS/TWAS/MAGMA and mendelian genes Speos could identify, the usefulness of method is very limited.

AUTHOR RESPONSE

As pointed out in our answer to 2.1a, the candidate genes are sourced from the unlabeled genes and therefore cannot contain the Mendelian genes. We have further removed all GWAS genes from the candidates and re-ran our validations, as requested by the reviewer. The mean and the upper and lower bounds of the confidence intervals of candidates are slightly lower when GWAS genes are removed, but the significance of the candidates predicted by the FiLM and TAG ensembles remains stable (see new Supplementary Fig. SF5a & b below). Additionally, the odds ratios for differentially expressed genes drop slightly for some diseases, such as for the subtypes of Immune Dysregulation, while remaining mostly stable for others (Supplementary Fig. SF5c). However, the main trend that candidate genes are strongly enriched for several core gene characteristics remains consistent, even after removing all GWAS genes from the candidates. We therefore conclude that most of the enrichment of candidate genes is not driven by GWAS genes and that Speos adds substantial value.

We have furthermore amended the last paragraph of the discussion, mentioning these analyses:

EXCERPTS FROM REVISED MANUSCRIPT

Results, Differential Gene Expression:

“Again, GWAS genes among candidates are not the main drivers of enrichment (**Supplementary Fig. SF5, Supplementary Tables 10-13**).”

Discussion:

“The validation results are robust, even when known GWAS genes are removed from the candidates, highlighting the added value of integrating multiple data modalities (**Supplementary Figures SF3 & SF5, Supplementary Tables ST3, ST10-ST13**).”

Supplementary Figure SF5 | External Validation with GWAS genes removed from

Candidates. a, b, LoF intolerance and missense mutation intolerance Z-scores of Mendelian genes, and the indicated candidate and non-candidate sets generated by the five methods. Shown are group means and 95% confidence intervals of Tukey's HSD test. Colored symbols and error bars indicate $FDR < 0.05$ in comparison with respective non-candidate sets; not significant sets in gray. Dashed line indicates the mean across all genes. **c**, Odds ratios (ORs) of Mendelian genes (first row) and of candidate genes of the five selected methods (rows) for common complex subtypes of the Mendelian disorder subgroups. ORs with $FDR > 0.05$ (Fisher's exact test) in gray. **d**, Enrichment of drug targets and druggability in Mendelian disorder genes and indicated candidate gene sets. DT: OR of known drug targets. xDC: Ratio of median number of drug-gene interactions per candidate gene to the median of non-candidates, only genes with drug-gene interactions are considered. Ratios with $FDR > 0.05$ (U-test) are grayed out. Dr: OR of druggable genes. Dr-: OR of druggable genes, after all drug targets have been removed. Odds Ratios with $FDR > 0.05$ (Fisher's exact test) are grayed out. Precise P-values, FDR, and n for each test in each panel are shown in **Supplementary Tables ST10 - ST13**, respectively.

2.2a Core genes are defined as genes that have direct effect on a disease, and it is only defined in a disease phenotype context. What is the rationale for grouping complex diseases into the five complex traits groups?

AUTHOR RESPONSE

In response to this and the following comments, we have added more information on the definition of Mendelian gene sets and their mapping to complex traits to the manuscript and we provide a short summary in the following. Indeed, core genes are only defined in the context of a disease phenotype. To map complex and rare Mendelian forms of a disease phenotype to each other, we follow the approach proposed by Freund et al. AJHG 2018, which is based on the clinical symptoms of a complex disease. Specifically, Freund et al. decompose complex diseases into a set of standardized clinical phenotype terms ("symptoms", see Figure below and **Supplementary Table ST26**). Each complex disease thus corresponds to a set of such symptoms and these can be used to identify Mendelian genes, mutation of which causes the same standardized clinical phenotypes. Just as a complex disease is not fully characterized by only one but many concurrent symptoms, this approach identifies a set of Mendelian genes that together characterizes the same complex disease phenotype.

Step 1: Query Mendelian genes using standardized clinical phenotype terms for 62 complex traits

Step 2: Hierarchical clustering of obtained gene sets based on threshold reveals 20 complex disease clusters

Step 3: Comparison of Mendelian disorder genes with genes solely sourced from significant GWAS hits, addition of sign. mappings

Visualization of the workflow proposed by Freund et al. To 1) query Mendelian disorder genes using lists of standardized clinical phenotype terms, 2) aggregate these gene lists to disease cluster using hierarchical clustering and 3) map GWAS genes of the complex disease counterparts to the disease clusters.

In step 2) of their analysis Freund et al. compared the Mendelian gene sets of several complex traits to each other using hierarchical clustering and found highly significant overlap between gene sets that correspond to highly related complex traits, which motivated them to merge these Mendelian gene sets into clusters. These overlaps are caused by pleiotropic effects of Mendelian genes that cause symptoms that occur in multiple related complex diseases. We are using these clusters of Mendelian disorder genes that correspond to the symptoms of highly related complex traits as positive labels for our model. An analogous observation has been made by ref. ¹⁰, who found shared genes among related diseases, and used this to generate a network of human diseases.

Finally, in step 3) they assessed the overlap between the Mendelian gene sets and gene sets representing the common form of a complex disease defined solely based on GWAS. They did this in a systematic way, comparing all common and Mendelian gene sets. Their key result was significant overlap of these gene sets occurs when the disease phenotypes are highly related (share many symptoms). Additionally, GWAS traits for complex diseases which only partially overlap with the Mendelian disease groups defined in step 2) are added to the mapping if they have a significant overlap in step 3 (i.e., RBC (red blood cell count) and BW (birthweight) for cardiovascular disease). This demonstrates that not only the complex trait that was used to define the corresponding Mendelian gene set, but also additional related complex traits carry information that can be predictive for Mendelian genes of the defining complex trait. Thus, grouping diseases with related clinical presentations increases not only statistical power, but also the ability to identify related patterns of symptoms and distinguish these from unrelated diseases.

In this sense our models are specific for groups of highly related complex diseases, as they are trained using labels that are based on the corresponding clinical symptoms. The models are trained to extract predictive patterns for the genes from molecular network, expression and GWAS data. Here, only features that increase the predictive performance receive high weights. On one

hand, if the GWAS of related complex traits carry predictive information, it will lead to improved predictions. If, on the other hand, only the complex trait that was used to define the corresponding Mendelian disorder gene set is relevant for the prediction of core genes, the GWAS features of any related complex traits should be downweighted accordingly during training. In both cases the predictions will be specific for the specific symptoms of related complex disease phenotypes, because the training objective is to recover the Mendelian disorder gene set corresponding to these symptoms. In conclusion, the rationale for grouping of multiple complex traits is to provide model with the largest gene set for training based on relevant symptoms. Including GWAS for related complex traits is thereby expected to improve prediction performance compared to predictions for any single complex disease.

To verify that no overfitting takes place and that indeed predictions using GWAS data of multiple related traits as input are as least as good as predictions based only on a single complex trait, we compared the results for cardiovascular disease to the predictions only for coronary artery disease (CAD). Most methods perform slightly worse than when all significantly associated traits are used as input (see Fig. below, a). Only TAG performs slightly better, which can be attributed to the biasing effect of the IntAct Direct Interaction network which likely has a stronger influence on the prediction as the input features now contain less information. We have discussed this bias and its effects in Supplementary Note 4. Additionally, the predicted candidate genes show a very high overlap with the candidate gene sets predicted by the model that uses all significantly associated traits for cardiovascular disease (see Fig. below, b). Only the MLP shows a weaker overlap, which is expected for a method that only relies on input features. However, MLP generally shows mixed performance across all validations. Based on the validation with mouse KO phenotypes, we conclude that no overfitting took place and the high overlap between the core genes predicted by using a single or by using multiple GWAS as input demonstrates that the predictions are highly concordant.

We have furthermore amended the Methods section with more details how the ground truth positives are assembled:

EXCERPT FROM MANUSCRIPT:

“Freund *et al.*⁴⁷ have recently defined 20 classes of Mendelian disorders which resemble common complex diseases. First, they defined lists of standardized clinical phenotype terms for 62 complex traits and used these lists to query the Online Mendelian Inheritance in Men (OMIM)⁹⁷ database to retrieve lists of Mendelian disorder genes for every complex trait. Subsequent hierarchical clustering of the retrieved gene lists reveals 20 disease group clusters among the 62 complex traits. Next, Freund et al gathered GWAS genes for the same 62 traits and compared the GWAS genes with the 20 Mendelian disease clusters. They observed that the (common complex) GWAS genes have a significant overlap with phenotypically related Mendelian disease clusters and used this significant association to map GWAS traits to the Mendelian disease clusters. Effectively this establishes a genetic and a symptom based connection between the Mendelian and the common complex forms of diseases”

2.2b When grouping complex traits into complex trait groups, I find it hard to define core genes. Are core genes specific to only one of the complex traits of the group defined as a core gene to the group? Clear definition of core genes is needed in the paper.

AUTHOR RESPONSE

We have adopted the definition of core genes from the omnigenic model, where they are defined as “the (minimal) set of genes such that, conditional on the genotype and expression levels of all core genes, the genotypes and expression levels of peripheral genes no longer matter”¹². This definition still holds true for each disease. However, for the reasons detailed above, we chose to group related complex diseases to increase the statistical power and the ability to reliably identify related patterns of clinical presentations. To better explain this methodological approach, we have updated and further specified the general definition of core genes in our introduction.

EXCERPT FROM MANUSCRIPT:

“The recently proposed “omnigenic” model postulates that the effects of genetic variation on a trait are mediated by core genes, encoding core proteins (hereafter used interchangeably depending on context), whose expression, and ultimately function, directly and mechanistically influences the phenotype, whereas peripheral genes and proteins constitute a regulatory network that propagates the effects of genetic variants on the phenotype by modulating core gene expression and function^{10,11}.”

AUTHOR RESPONSE (CONTINUED)

As detailed in our answer to 2.2.a, the complex trait groups have been defined based on a symptom-based query to capture the shared underlying mechanisms between closely related diseases. To provide a specific example: Immune dysregulation includes several complex diseases characterized by a pathological response of the immune system, with the main discerning factor being the affected tissue. We therefore find it plausible that a gene that has a direct mechanistic connection and strong potential to dysregulate the immune system, i.e. a core

gene for immune dysregulation, can be implicated in several common complex subforms. Examples of this behavior are IL-6 and the genes encoding the two subunits of NF-kappa-B, which are among the Mendelian disorder genes and are used as positive examples during training. We have, however, noticed that some of the predicted candidates seem to be more relevant to some of the subforms than to others. The two selected candidates for immune dysregulation, for example, seem to show a stronger connection to Crohn's disease, inflammatory bowel disease and celiac disease than to systemic lupus erythematosus (**Figure 5, c & d**). In a similar manner, the two selected candidates for cardiovascular disease seem to be most relevant to dilated cardiomyopathy, as their expression in the left ventricle and atrial appendage are the most important features (**Extended Data Figure 10, c & d**). Finally, Figure 4a shows that the predicted candidate gene sets of the best methods, FiLM and TAG, capture most subforms of the complex diseases indicating that, while individual candidates might be more subtype-specific, the whole set captures the underlying similarities.

In addition, in response to this and the previous comments, we have now added a more precise definition of how core genes for specific traits are identified in our work to the main text (changes in italic).

EXCERPT FROM MANUSCRIPT:

“As Mendelian genes display all characteristics of ‘strong’ core genes¹⁴ we use these as positive labels for a positive unlabeled graph representation learning²⁷ ensemble (Supplementary Fig. SF1). To specify the Mendelian genes corresponding to specific complex diseases, we make use of the mapping established by Freund et al.⁴⁷. It uses standardized clinical phenotype terms that characterize the specific symptoms of complex traits to identify sets of Mendelian genes and groups closely related diseases into Mendelian disorders clusters that are mapped to closely related complex diseases. Thus, predicted core genes are specific for these groups of highly related diseases, as the models are trained using labels that are defined by disease specific standardized phenotype terms.”

Discussion:

*“In summary, we show that Speos is able to produce candidate core gene sets for different common and complex diseases using Mendelian disorder genes as training examples (**Supplementary Table ST24**). We used a systematic mapping of complex traits to Mendelian genes⁴⁷ as input to demonstrate the general power of the method on several diseases. More fine-grained analyses are supported by the framework and can easily be implemented for specific traits of interest by specifying the Mendelian gene sets.”*

2.2c In addition, I also failed to see the disease specificity of core genes in the mouse knockout analyses. Would the core genes in one disease group have increased OR in the rest of the four disease groups?

AUTHOR RESPONSE

The mouse knockout validation is disease specific as it uses the same standardized clinical phenotype term queries to obtain each diseases' mouse knockout genes (see **Supplementary Table ST26**) that have been used by Freund et al.¹¹ to obtain the Mendelian genes. Naturally, given the semantic nature of the queries and the pleiotropy of some genes, some overlap between the retrieved gene sets is to be expected. As an obvious example, nearly all diseases involve some degree of inflammation and immune response. To ascertain the disease specificity of the mouse knockout validation, we have validated each group's candidates and Mendelian genes against the mouse knockout genes of immune dysregulation (see Supplementary Fig. SF4). In general, cross-disease evaluations of candidates and Mendelian genes yield low odds ratios at best, with some methods such as GCN, N2V+MLP failing to reach any enrichment, and some (MLP candidates for cardiovascular disease) even showing depletion. Lower consensus score bins of candidates predicted by FiLM and TAG still contain a significant enrichment while higher bins fail to do so. Thus, we conclude that these classifiers capture the similarities between the diseases in the lower bins, while higher, potentially more disease-specific bins, are only enriched in the correct mapping (i.e., immune dysregulation), effectively demonstrating disease group specificity.

Supplementary Figure SF4 | Mouse Knockout validation disease-specificity experiments.

Candidate genes for all five disorders are validated against the mouse knockout genes for immune dysregulation. Odds ratio (OR) (right y-axis) for observing disease relevant phenotypes in mice with knockouts of orthologs of candidate core genes in the indicated convergence score bins (x-axis) of the five classifier methods (colored lines). Gray lines indicate strength of candidate gene sets (left y-axis) in the corresponding bin for the phenotypes as indicated in the panel. Only ORs with an FDR < 0.05 (Fisher's exact test) are shown. Bars to the right (M) and left (G) of each

plot indicate set strength (gray) and OR (colored) of Mendelian genes and GWAS genes for each phenotype. Filled bars represent ORs with an FDR < 0.05, otherwise bars are hollow. Precise P-values, FDR, and n for each test are shown in **Supplementary Table ST5**.

However, during preparation of this analysis, we noticed that the mouse knockout genes for insulin disorders in Fig. 3a used the wrong query. We have updated **Figure 3a, Extended Data Fig. 9a** and the accompanying tables **Supplementary Table ST3, ST5, ST24** as well as Supplementary Table **ST7**, now using the correct query.

2.2d The definition of the disease groups is also strange such that some diseases are in two different groups, for example, BMI is in two groups. It seems rather random and no clear explanation were given in the manuscript

AUTHOR RESPONSE

As detailed above (2.2a), disease groups are defined by standardized clinical phenotype terms established by Freund et al. AJHG 2019 and based on statistics of shared underlying genes. The fact that individual complex traits occur in multiple disease groups is not random but explained by step 3) of the classification approach, which assessed the overlap between the Mendelian gene sets and gene sets representing the common form of a complex disease defined based on systematically collected GWAS data. This was done in a systematic way, comparing all common and Mendelian gene sets. A key result was that significant overlap of these gene sets occurs when the disease phenotypes are highly related, i.e. share many symptoms. This demonstrates that not only the complex trait that was used to define the corresponding Mendelian gene set, but also additional related complex traits carry information that can be predictive for Mendelian genes of the defining complex trait. In the case of BMI this means that genetic causes underlying the complex trait BMI on one hand and the Mendelian disorder groups ‘body mass disorders’ and ‘insulin disorders’ are overlapping. Thus, we expect that BMI GWAS data can be predictive for core genes of both Mendelian disorder groups. The phenotypic connection of BMI and ‘body mass disorders’ is evident, while ‘insulin disorders’ includes Mendelian genes related to ‘Insulin Resistance & Fasting Glucose’, which is often observed as part of the metabolic syndrome that among other traits is characterized by obesity, again clearly related to BMI.

As suggested by the reviewer we have added an explanation to the main text (changes italic).

EXCERPT FROM MANUSCRIPT:

“As Mendelian genes display all characteristics of ‘strong’ core genes¹⁴ we use these as positive labels for a positive unlabeled graph representation learning²⁷ ensemble (**Supplementary Fig. SF1**). *To specify the Mendelian genes corresponding to specific complex diseases, we make use of the mapping established by Freund et al.⁴⁷. It uses standardized clinical phenotype terms that characterize the specific symptoms of complex traits to identify sets of Mendelian genes and groups closely related diseases into Mendelian disorders clusters that are mapped to closely*

related complex diseases. Thus, predicted core genes are specific for these groups of highly related diseases, as the models are trained using labels that are defined by disease specific standardized phenotype terms."

Moreover we have amended the methods section (changes in italic).

EXCERPT FROM MANUSCRIPT:

"Freund et al.⁴⁷ have recently defined 20 classes of Mendelian disorders which resemble common complex diseases. First, they defined lists of standardized clinical phenotype terms for 62 complex traits and used these lists to query the Online Mendelian Inheritance in Men (OMIM)⁹⁷ database to retrieve lists of Mendelian disorder genes for every complex trait. Subsequent hierarchical clustering of the retrieved gene lists reveals 20 disease group clusters among the 62 complex traits. Next, Freund et al gathered GWAS genes for the same 62 traits and compared the GWAS genes with the 20 Mendelian disease clusters. They observed that the (common complex) GWAS genes have a significant overlap with phenotypically related Mendelian disease clusters and used this significant association to map GWAS traits to the Mendelian disease clusters. Effectively this establishes a genetic and a symptom based connection between the Mendelian and the common complex forms of diseases.

2.3a Results for cardiovascular diseases seems to be an outlier, especially the LOF and missense analyses in Figure 4b and 4c. Simply stating that "...indicating a potentially interesting, but at this point unexplained phenomenon." is not enough. Did the model simply fail?

AUTHOR RESPONSE

The reviewer's comment prompted us to further investigate this topic. First, we want to clarify that this concerns only one classifier (FiLM) and that the same candidate genes that display a significant depletion of loss of function (LoF) intolerance show results in line with our understanding of a core gene in every other validation (see results for FiLM in **Fig. 3, 4a, 6**). Therefore, the overarching characteristics of the core gene candidates proposed by the FiLM model are still mostly in line with our understanding of core genes, despite the unexpected result from the LoF intolerance validation.

We have added our novel analyses in Supplementary Note 5:

"Upon closer examination, the query which leads to the Mendelian genes for cardiovascular disease mixes two distinct phenotypes: 1) a phenotype that is more focused on the heart muscle, its insufficiency or anatomical anomalies and 2) a phenotype that is mostly focused on the coronary arteries, stenosis and the cumulative deposits of plaque consecutively leading to disease. We therefore hypothesized that the unexpected results in the LoF validation are rooted in the classifier's inability to fully capture these two distinct phenotypes. To test this hypothesis in an ablation experiment, we have split the query terms into two lists, 'heart disease' and 'coronary artery disease', to more precisely capture the two subtypes. We used these queries to obtain new sets of Mendelian genes from OMIM, which we conditioned to be strict subsets of the Mendelian

genes previously used for cardiovascular disease, to not dilute the meticulous pre-selection carried out by Freund *et al.*⁴⁷. The query for heart disease returned 400 Mendelian disorder genes while the query for coronary artery disease returned 271 Mendelian disorder genes with an overlap between the two sets of size 134 (see **Supplementary Table ST21** for queries and retrieved genes). We used both lists to train new ensembles and repeated the LoF validation with the resulting candidate genes. Once the two phenotypes are separated, the LoF validation turns out more in line with our understanding of core genes (see new **Supplementary Fig. SF6a & b below**), while the rest of validations still return strong results (**Supplementary Fig. SF6c-e**). We therefore conclude that our hypothesis is plausible; it appears that, in this instance, mixing of two distinct phenotypes caused the unexpected result. The newly introduced depletion in the results of the GCN classifier can be attributed to its generally mixed performance in several validations.”

Please note that, to allow for comparison with our initial results, we only modify one component of the prediction task at a time. This ablation study uses subsets of the labels while in the ablation study shown in response to 2.2a we have used subsets of input GWAS features, thus both ablation studies can be compared to the initial results in the manuscript, but not directly to each other.

These results indicate that more specific phenotypes can be extracted from the ones currently implemented. Nonetheless, we consider it important to keep our Mendelian disorder gene sets congruent with Freund *et al.* because we can confidently redefine cardiovascular queries based on our expertise in this area, but redefining query sets for all disease groups is beyond the scope of this work. We believe that the well-documented source code will enable user groups with a specific interest in and in-depth knowledge of particular disease groups to refine the queries and thus take advantage of the developed framework. Additionally, we have made the following amendments to the respective section in the main text (changes in italic).

EXCERPT FROM MANUSCRIPT:

“For cardiovascular diseases, the FiLM predictions again show a significant depletion indicating a *potential heterogeneity in the definition of the cardiovascular disease phenotype (Supplementary Note 5, Supplementary Figure SF6, Supplementary Tables ST17-21).*”

Supplementary Figure SF6 | External Validation of Novel Phenotypes. **a, b**, LoF intolerance and missense mutation intolerance Z-scores of Mendelian genes, and the indicated candidate and non-candidate sets generated by the five methods. Shown are group means and 95% confidence intervals of Tukey's HSD test. Colored symbols and error bars indicate $P < 0.05$ in comparison with respective non-candidate sets; not significant sets in gray. Dashed line indicates the mean across all genes. **c**, Odds ratio (OR) (right y-axis) for observing disease relevant phenotypes in mice with knockouts of orthologs of candidate core genes in the indicated convergence score bins (x-axis) of the five classifier methods (colored lines). Gray lines indicate strength of candidate gene sets (left y-axis) in the corresponding bin for the phenotypes as indicated in the panel. Only ORs with an FDR < 0.05 (Fisher's exact test) are shown. Bars to the

right (M) and left (G) of each plot indicate set strength (gray) and OR (colored) of Mendelian genes and GWAS genes for each phenotype. Bars representing significant ORs are filled, hollow bars represent non-significant ORs. **d**, Odds ratios (ORs) of Mendelian genes (first row) and of candidate genes of the five selected methods (rows) for common complex subtypes of the Mendelian disorder subgroups. ORs with FDR > 0.05 (Fisher's exact test) in gray. **e**, Enrichment of drug targets and druggability in Mendelian disorder genes and indicated candidate gene sets. DT: OR of known drug targets. xDC: Ratio of median number of drug-gene interactions per candidate gene to the median of non-candidates, only genes with drug-gene interactions are considered. Ratios with FDR > 0.05 (U-test) are grayed out. Dr: OR of druggable genes. Dr-: OR of druggable genes, after all drug targets have been removed. Odds Ratios with FDR > 0.05 (Fisher's exact test) are grayed out. Precise P-values, FDR, and n for each test in each panel are shown in **Supplementary Tables ST16 – ST20**, respectively.

2.3b Are the well-known cardiovascular disease core genes such as APOE or LDLR identified? What are their CS? I don't have confidence in the results, unless the authors could show some evidence that the results make sense. It might also benefit the paper if the authors could go deeper into the cardiovascular disease results.

AUTHOR RESPONSE

The reviewer has pointed out two important cardiovascular disease core genes, APOE and LDLR. However, both genes mentioned, along with LDLRAP1 and several other genes coding for relevant apolipoproteins are among the already known Mendelian genes¹¹ and therefore cannot be predicted as novel core genes. Since they are used as positives during training, they do not receive a consensus score and cannot be selected as candidate genes in Speos' cross-validation ensemble. However, by design of the data splits in the ensemble, each positive gene is partitioned to the test set in one of the outer folds, i.e. is held out for all ten inner folds of the respective outer fold. We can use the raw predictions and compare them to the respective inner threshold, i.e., if the gene would have been unlabeled (unknown), would it have been selected as candidate by this outer fold. APOE surpasses the inner threshold for every method we used, while LDLR surpasses it for 4 out of 5 methods. If the genes would have been unlabeled, there would have been 11 chances (outer folds) for them to be predicted as a candidate and not only one. So, the fact that the one fold where they are partitioned to the test set has a 90% rate of predicting them is a strong indication that the genes would be among the candidates if they would have been unlabeled (unknown).

2.4 The main text shows the AUROC curve with reasonable numbers, however the AUPRC in Supplement figure 3 is very low (all < 0.15). Core genes should be a small number in comparison

to genome wide genes. Is AUPRC better suited than AUROC in this case? Does the low AUPRC mean the predictive value (probability of true positive given positive test) of the model is very low? If so, it is concerning.

AUTHOR RESPONSE

We thank the author for bringing up this important distinction between the performance metrics. However, it is crucial to understand that these performance metrics are only useful for method selection and to gain an insight into which training modalities (i.e., methods, input data etc.) are helpful for identifying held out core genes. They are explicitly not useful in extrapolating the performance to discovering new core genes, which is the objective of our study, as these genes are, by definition, unlabeled (unknown) and thus cannot be validated using holdout sets. It is therefore important to distinguish the performance of a certain base classifier in discovering held out Mendelian genes (AUROC / AUPRC) and the ability of a cross-validation ensemble to predict novel core genes (external validation metrics such as mouse KO data). We have strictly separated these two steps and made it clear that AUROC/AUPRC is just used to select suitable training modalities (**Supplementary Figure SF1c**). In this case, the absolute performance measured by AUROC and AUPRC is secondary, as we select the methods that have the highest performance relative to the other methods.

The low AUPRC values are a consequence of the rare positive labels in PU learning. Since we assume negative labels for all unlabeled genes during training, “hidden” or “previously unknown positive” core genes that rightfully receive a high prediction will be counted as a misclassification during the calculation of AUROC and AUPRC (all previously unknown positives are naturally considered as false positives in this calculation). Since this behavior severely influences the precision metric, AUPRC has only limited value in PU scenarios. Several sources argue that precision, and therefore AUPRC, is entirely unsuitable in PU scenarios because the rate of false positives cannot be accurately quantified due to the “hidden” positives^{13,14}. In other approaches that are based on a semi-supervised learning approach instead of PU learning, one can calculate a more faithful AUPRC using the negative labels². However, these approaches usually choose trivial negative examples and exclude the unlabeled genes when calculating their AUPRC, which therefore often give rise to artificially inflated metrics. To provide the reader with a more well-rounded perspective on the model performance we have chosen to include the AUPRC tables in the supplementary and only show AUROC in the main part of the manuscript, alongside with an explanation why AUROC is still suitable in our scenario:

“AUROC is suitable for model comparison in PU learning as known positives receive higher predictions than the average unlabeled gene, even though the unlabeled (actual) positives reduce the optimal AUROC score.”

2.5 Relevant to my previous comment, the number of candidate core genes is too big (Figure 3a). Looks like all trait groups have 1000s of core genes, which is impossible. Combining with the low AUROC, does it mean most of the candidate genes are not true core genes?

AUTHOR RESPONSE

As we have detailed in our response to remark 2.4, the magnitude of AUROC values of our base classifiers is expected in a positive unlabeled setting and must not be conflated with the AUROC of related work, which mostly uses the semi-supervised learning setting. Importantly, the validity of the predicted core genes is established by the external validation analyses. Regarding the number of core gene candidates, the original authors of the omnigenic model do not give any hint as to how many genes can be expected to be core genes, only that the core genes are “the (minimal) set of genes such that, conditional on the genotype and expression levels of all core genes, the genotypes and expression levels of peripheral genes no longer matter”¹². Looking at the Mendelian disease gene sets for comparison, we find that, while some sets are very concise with only slightly more than 100 genes, others contain more than 500 genes¹⁰. Given that the discovery of a Mendelian mutation and hence a Mendelian gene also depends on random factors (has a family carrying a rare mutation in that gene been studied or not), we tend to think of them like the “tip of the iceberg”, with more core gene candidates hidden underneath the surface. Since we are working with complex diseases which encompass dozens of tissues and cell types and possibly hundreds of regulatory pathways, it is not unfathomable for us to arrive at a number in the low 1000s for the most permissive setting (i.e Consensus Score ≥ 1). Most crucially, most of our external validations are carried out with these large sets of core gene candidates (Fig. 4 & 6) and show a strong, significant signal in line with our understanding of core genes. It is very unlikely to observe such strong signal if most of the candidates were not real core genes. However, if more concise sets should be required, e.g., for a downstream application that requires large resources per gene, our results suggest that the more stringent sets obtained by imposing a higher Consensus Score cutoff display an even stronger signal of core genes (Fig. 3). Since we cannot judge the requirements regarding the tradeoff between sensitivity and specificity of potential downstream use cases, we have decided to publish our candidate genes alongside their Consensus Score at the user's discretion (see **Supplementary Table ST24**).

2.6 (Minor comment) Overall, I think the writing is generally clear. However, the intuition behind the complicated method can be better explained in the main text.

AUTHOR RESPONSE

To give a better intuition of the overall workflow of the method, we have added an overview figure (new Supplementary figure SF1 shown below). In addition, we have also added the motivation for applying the ensemble approach to the main text (changes in *italic*).

EXCERPT FROM MANUSCRIPT:

The next question in PU learning, is how to decide on a suitable threshold for the prediction of a novel previously unknown core gene. To address this question, we propose a statistical approach to select thresholds based on nested cross-validation. For this ensemble method, we selected five methods as base classifiers, based on their performance during method selection: N2V+MLP, which had the best overall performance, FiLM trained on all networks, and TAG trained on IntAct Direct Interaction as best performing GNN-based methods. Despite the lower performance we decided to also include MLP as a baseline classifier that does not use relational network information, and GCN48, which is regularly used in graph-based problems to ensure comparability with other studies.

Supplementary Figure SF1 | Graphical Overview. **a:** The main objective of the study is to predict novel core genes for common complex diseases using Mendelian disorder genes as ground truth “strong” core genes according to the omnigenic model. **b:** We integrate multi-modal data either as feature vectors or as network data. A feature vector containing GWAS and gene expression features is assigned to every gene, genes with missing data are discarded. Genes are connected by several different biological networks which can be undirected or directed. Each network can be used individually or in conjunction with other networks. **c:** Hyperparameter search testing for various variations of networks, input data and methods to identify the method and setting that is best suited to recover held out Mendelian disorder genes. **d:** Development of a nested crossvalidation ensemble that leverages consensus between models in a two-stage process to predict novel core gene candidates from unlabeled genes. The best performing classifiers from step **c** are used as base classifiers for the ensemble. **e:** The core gene candidates identified in step **d** are selected for external validation by testing for several criteria that are

expected for core genes, such as disease-specificity in knockout experiments and differential expression when the disease is present.

Literature Cited in Response Letter

1. Kipf, T. N. & Welling, M. Semi-Supervised Classification with Graph Convolutional Networks. *ArXiv160902907 Cs Stat* (2016).
2. Schulte-Sasse, R., Budach, S., Hnisz, D. & Marsico, A. Integration of multiomics data with graph convolutional networks to identify new cancer genes and their associated molecular mechanisms. *Nat. Mach. Intell.* **3**, 513–526 (2021).
3. Grover, A. & Leskovec, J. node2vec: Scalable Feature Learning for Networks. in *Proceedings of the 22nd ACM SIGKDD International Conference on Knowledge Discovery and Data Mining - KDD '16* 855–864 (ACM Press, 2016). doi:10.1145/2939672.2939754.
4. Brockschmidt, M. GNN-FiLM: Graph Neural Networks with Feature-wise Linear Modulation. Preprint at <http://arxiv.org/abs/1906.12192> (2020).
5. Du, J., Zhang, S., Wu, G., Moura, J. M. F. & Kar, S. Topology Adaptive Graph Convolutional Networks. Preprint at <http://arxiv.org/abs/1710.10370> (2018).
6. Leeuw, C. A. de, Mooij, J. M., Heskes, T. & Posthuma, D. MAGMA: Generalized Gene-Set Analysis of GWAS Data. *PLOS Comput. Biol.* **11**, e1004219 (2015).
7. Watanabe, K., Taskesen, E., Bochoven, A. van, Posthuma, D. Functional mapping and annotation of genetic associations with FUMA. *Nat. Comm.* **8**, 1826 (2017)
8. Cacheiro, P. *et al.* Human and mouse essentiality screens as a resource for disease gene discovery. *Nat. Commun.* **11**, 655 (2020).
9. Bishop, C. M. *Pattern Recognition and Machine Learning*. New York, Springer, 2006, p. 653
10. Goh, K. *et al.* The human diseases enetwork. *PNAS Appl Phys Sc*, **104** (21) 8685-8690, 2007
11. Freund, M. K. *et al.* Phenotype-Specific Enrichment of Mendelian Disorder Genes near GWAS Regions across 62 Complex Traits. *Am. J. Hum. Genet.* **103**, 535–552 (2018).
12. Boyle, E. A., Li, Y. I. & Pritchard, J. K. The Omnigenic Model: Response from the Authors. *J. Psychiatry Brain Sci.* **2**, (2017).
13. Bekker, J. & Davis, J. Learning from positive and unlabeled data: a survey. *Mach. Learn.* **109**, 719–760 (2020).

14. Claesen, M., Davis, J., Smet, F. de, Moor, Bart de. Assessing binary classifiers using only positive and unlabeled data. Preprint at <https://arxiv.org/abs/1504.06837> (2015).

REVIEWERS' COMMENTS

Reviewer #1 (Remarks to the Author):

The authors have adequately addressed all of the concerns raised during the revision.

Reviewer #2 (Remarks to the Author):

The authors have addressed my comments 2.1, 2.3 and 2.4 with additional analyses and rewriting. However, I am not satisfied with the answers for 2.2 and 2.5, mostly because of the use and misuse of “core genes”.

Core genes, “directly and mechanistically influence the phenotype”(quote from their main text) and it is disease specific. “Core genes” cannot be defined for a group of diseases, which is the way the authors use it.

In addition, while extreme core genes have properties of Mendelian genes, and the authors use Mendelian genes as training for their model, the authors seemed to confuse themselves between core genes and Mendelian genes. For example, in their argument about the number of core genes, they were using the number of Mendelian genes to represent “core genes”. Not all Mendelian genes relevant to a trait are core genes.

Core genes are also “the (minimal) set of genes such that, conditional on the genotype and expression levels of all core genes, the genotypes and expression levels of peripheral genes no longer matter”

Throughout the paper, these properties of the core genes were not used or verified. Their mouse knockout and all validations are only validating if the predicted gene have mendelian properties or have effect on the phenotype. Their candidate genes were not about “core genes”, but rather just “disease genes” with Mendelian properties.

The use of “core genes” as is in the manuscript is a misuse and extremely misleading to the readers. Without careful use of the term, the users of their method will think they are predicting “core genes” in their own study and further exacerbate the misuse of the term. Therefore, I strongly think that the authors should

1) change their title to be more accurate and less misleading, something like: Speos: An ensemble graph representation learning framework to predict important disease genes for complex disease groups

2) update their text, where appropriate, to use more accurate term such as “important disease genes with Mendelian properties”, rather than “core genes”. For example, line 102-103: Here we present Speos,to predict “important” gene candidates with Mendelian properties for five groups of complex diseases.

The predicted core genes -> the predicted disease genes

Core genes of cardiovascular disease-> important genes of cardiovascular diseases

And many more.

Additionally, the authors should carefully check their revised manuscript that it contains all the revision they mentioned in rebuttal. For example, the following revision mentioned in the rebuttal was not found in the manuscript (at least in the version I see).

“Within each outer fold we statistically assess the agreement of the predictions for the outer holdout (test) set and the unlabeled genes of the 10 inner models. Using the outer fold hold-out set, we select an inner threshold at which the agreement among the 10 inner models on held out Mendelian genes surpasses random expectation ($FDR < 0.05$; Student’s t-test, Fig. 2b, Supplementary Table ST1). All unlabeled genes with higher agreement than this inner threshold are considered candidate genes of this outer fold. Mendelian genes cannot be selected as candidates, as they are already known positives and predictions are only computed for unlabeled genes.”

We would like to thank the reviewers for their time and feedback on our revised manuscript. We are addressing the remaining critiques and concerns below.

1. The authors have addressed my comments 2.1, 2.3 and 2.4 with additional analyses and rewriting. However, I am not satisfied with the answers for 2.2 and 2.5, mostly because of the use and misuse of “core genes”. Core genes, “directly and mechanistically influence the phenotype” (quote from their main text) and it is disease specific. “Core genes” cannot be defined for a group of diseases, which is the way the authors use it.

AUTHORS RESPONSE

Upon considering the reviewer’s continued dissatisfaction we realized that, indeed, the (matter factly) term “core gene” is not appropriate without strong and direct experimental support. Therefore, we now agree that the term “core-gene” for predicted genes is not optimal and now refer to them as “core-like genes”.

Providing definitive proof for individual genes to be causal “core genes” (in the sense of the omnigenic model) would require extensive genetic manipulation and observation of resulting phenotypes in humans, which is not feasible foremost due to ethical reasons. Therefore, only indirect evidence from mouse knockout data (causality in mouse), human genetics (negative selection) and differential expression in disease (association) are available, which are all consistent with the properties of “core genes” that are postulated by the omnigenic model. To reflect that our predicted genes have properties of core genes, but lack final proof, we now refer to them as “core-like genes”. We have revised the manuscript to reflect this point:

EXCERPT FROM MANUSCRIPT:

Abstract:

“Using external validations, we demonstrate that core-like genes display several key properties of core genes: Mouse knockouts of genes corresponding to our most confident predictions give rise to relevant mouse phenotypes at rates on par with the Mendelian disorder genes, and all candidates exhibit core gene properties like transcriptional deregulation in disease and loss-of-function intolerance.”

Introduction:

“Proving a gene to be a core gene for complex traits either requires unethical human genetic intervention studies or epidemiological human data from extremely large sample sizes¹⁴. Therefore, we refer to the predicted genes, which exhibit several expected properties of core genes, such as causality in mice, negative selection and differential disease expression in humans, as “core-like”.”

Discussion:

“Finally, the core-like gene sets predicted by Speos can be used to prioritize genes for experimental validation to provide more definitive evidence of being core genes.”

AUTHORS RESPONSE CONTINUED

The second point of the reviewer's argument is whether core-genes can be defined for disease groups, which the reviewer negates.

On this point we respectfully disagree in light of genetic disease-relationships and genetic pleiotropy. In addition to our initial arguments, which we would like to recall without repeating them, our thinking is backed by the following observations and data. The literature is abounded with genes that are causal in multiple related diseases: Among the complex diseases that the omnigenic model has been developed for and which we study, major histocompatibility complex, class II, DQ BETA-1; HLA-DQB1 is a Mendelian (core) gene for two autoimmune diseases (celiac disease and multiple sclerosis). Filamin A is a Mendelian gene associated with 10 different phenotypes. Similarly, among related diseases causal genes can be shared: Interleukin 23 levels are clinically and mechanistically relevant for several autoimmune diseases, such as psoriasis, rheumatoid arthritis, multiple sclerosis, systemic lupus erythematosus and inflammatory bowel disease (Molano-Gonzalez *et al.* J Autoimmun, 2019; Abdo *et al.*, Inflamm Res, 2020). IL23R, the gene encoding the interleukin 23 receptor, a key protein for interleukin 23 signaling, is among the Mendelian disorder genes for immune dysregulation and therefore clearly a core gene on disease group level.

Beyond such very specific examples, systems-level analyses demonstrate that common genes are underlying related diseases. Continuing with our example of immune dysregulation, a high genetic similarity between the traits is well recognized (Parkes *et al.*, Nat Rev Gen, 2013) and a large cluster of shared disease genes among autoimmune diseases has been known for more than a decade (Richard-Miceli and Criswell, Genome Med, 2012). This cluster has been continuously extended and now encompasses 18 autoimmune diseases and hundreds of genes (Golukhadas *et al.*, Front Immunol 2021). Clinically, this high degree of pleiotropy is mirrored in the fact that multiple autoimmune diseases often co-occur simultaneously in the same patient, a phenomenon described as polyautoimmunity (Ordoñez-Cañizares *et al.*, J Clin Rheumatol, 2020; Rojas *et al.*, J Autoimmun, 2022; Molano-Gonzalez *et al.* J Autoimmun, 2019). Even more general, a study from 2008 (Goh *et al.*, PNAS) systematically mapped the extent of disease relatedness based on common OMIM genes underlying these diseases (see also Freund *et al.*, Am J Hum Genet, 2018; Barrio-Hernandez *et al.*, Nat Gen, 2023). Also in this case, related diseases clustered together as they share both common and rare perturbations to the same underlying processes. Phenotypically this can be conceptualized by looking at symptoms. When diseases share specific symptoms or molecular phenotypes, which is how our disease groups were composed in the first place, it is likely that similar pathways and hence common genes and proteins are involved. Since the disease groups are defined by similarity in clinical phenotypes, a large overlap in the respective genes for every trait within the group is expected and observed (Freund *et al.*, Am J Hum Genet, 2018).

Thus, we do not agree with the notion that core genes cannot be common to closely related diseases.

Having said that, an important question regarding the set of core-like genes that we identify for each disease group is what proportion of those is common to all individual diseases, what

proportion is common to several diseases, and what fraction is specific to only one disease in the group. This is a question we are not able to answer at this stage. Our conclusions are on the level of "we predict IL23R is core gene for immune dysregulation" without being able to specify for which immune trait(s) or diseases. We have expanded the corresponding paragraph in the discussion to account for this lack of granularity.

EXCERPT FROM MANUSCRIPT:

"More fine-grained analyses are supported by the framework and can easily be implemented for specific traits of interest by specifying the Mendelian gene sets. **This will also allow for determining whether the predicted genes are only relevant for individual traits or show substantial pleiotropy within disease groups.** "

2. In addition, while extreme core genes have properties of Mendelian genes, and the authors use Mendelian genes as training for their model, the authors seemed to confuse themselves between core genes and Mendelian genes. For example, in their argument about the number of core genes, they were using the number of Mendelian genes to represent "core genes". Not all Mendelian genes relevant to a trait are core genes.

AUTHORS RESPONSE

This appears to be a fundamental misunderstanding. Indeed, Mendelian genes are considered core genes: "Mendelian disease clearly fulfills the core gene definition, as disease only occurs in the context of a given mutation." (Wray *et al.*, 2018, Cell). We adhered to this commonly accepted definition. Additional core genes, which are not already known Mendelian genes, are likely to exist. So, to estimate the number of *bona fide* core genes (stronger and weaker core genes) we take the number of known Mendelian disorder genes (strong core genes) into account as a lower bound.

3. Core genes are also "the (minimal) set of genes such that, conditional on the genotype and expression levels of all core genes, the genotypes and expression levels of peripheral genes no longer matter"

Throughout the paper, these properties of the core genes were not used or verified. Their mouse knockout and all validations are only validating if the predicted gene have mendelian properties or have effect on the phenotype. Their candidate genes were not about "core genes", but rather just "disease genes" with Mendelian properties.

AUTHORS RESPONSE

As indicated above, we have now adjusted our terminology from "core genes" to "core-like genes" to reflect the fact that formally proving causality (or conditional independence as indicated in the sentence cited by the reviewer) would require additional intervention experiments in humans, which are not feasible due to ethical reasons. Nevertheless, we would like to stress that we do provide evidence that predicted "core-like" genes are indeed causal for the phenotype in mice, where genetic intervention experiments (gene knock out) have been performed and indeed

demonstrate that mice develop related phenotypes more often than expected by chance, when predicted “core-like” genes are knocked out. This enrichment has a similar order of magnitude as that of Mendelian genes, which are well recognized to be core genes, as mentioned above.

4. The use of “core genes” as is in the manuscript is a misuse and extremely misleading to the readers. Without careful use of the term, the users of their method will think they are predicting “core genes” in their own study and further exacerbate the misuse of the term. Therefore, I strongly think that the authors should

1) change their title to be more accurate and less misleading, something like: Speos: An ensemble graph representation learning framework to predict important disease genes for complex disease groups

2) update their text, where appropriate, to use more accurate term such as “important disease genes with Mendelian properties”, rather than “core genes”. For example, line 102-103: Here we present Speos,to predict “important” gene candidates with Mendelian properties for five groups of complex diseases.

The predicted core genes -> the predicted disease genes

Core genes of cardiovascular disease-> important genes of cardiovascular diseases

And many more.

AUTHORS RESPONSE

We have now adjusted our terminology from “core genes” to “core-like genes”. We believe this makes it clear that the predicted genes are not to be taken as literal core genes but as candidates which require further empirical validation to be finally accepted as core genes for individual phenotypes.

At the same time, the entire Speos approach is based on the omnigenic model for gene function, in which the term ‘core gene’ is precisely defined (albeit perhaps different than how the term has been used historically). However, because ‘core gene’ is a specific and defined term in the model and because our approach is informed by the omnigenic model, a more drastic rephrasing of ‘core-like’ to ‘important’ would not add to the clarity of presentation. Similarly, we believe that the title is not the right place to introduce the difference between core genes and core-like genes, which is why we changed it to “core gene candidates”, which implies that the genes require further experimental validation.

Besides changing the term throughout the manuscript, we have also added sections to the abstract, introduction and discussion highlighting the distinction between the core genes as conceptualized and defined in the omnigenic model and specific core-like gene predicted by Speos:

EXCERPT FROM REVISED MANUSCRIPT:

Title: Speos: An ensemble graph representation learning framework to predict core gene **candidates** for complex diseases

Abstract:

“Using external validations, we demonstrate that core-like genes display several key properties of core genes: Mouse knockouts of genes corresponding to our most confident predictions give rise to relevant mouse phenotypes at rates on par with the Mendelian disorder genes, and all candidates exhibit core gene properties like transcriptional deregulation in disease and loss-of-function intolerance.”

Introduction:

“Proving a gene to be a core gene for complex traits either requires unethical human genetic intervention studies or epidemiological human data from extremely large sample sizes¹⁴. Therefore, we refer to the predicted genes, which exhibit several expected properties of core genes, such as causality in mice, negative selection and differential disease expression in humans, as “core-like”.”

Discussion:

“Finally, the core-like gene sets predicted by Speos can be used to prioritize genes for experimental validation to provide more definitive evidence of being core genes.”

References

1. Cluster analysis of autoimmune rheumatic diseases based on autoantibodies. New insights for polyautoimmunity. Molano-Gonzalez et al. *J Autoimmun* 98:24-32, 2019
2. Interleukin 23 and autoimmune diseases - current and possible future therapies. Abdo et al., *Inflam Res* **69**, 463-480, 2020
3. Genetic insights into common pathways and complex relationships among immune-mediated diseases. Parkes et al. *Nat Rev Gen* **14**: 661-673, 2013
4. Emerging patterns of genetic overlap across autoimmune disorders. Richard-Miceli C, Criswell LA. *Genome Medicine* **4** (6), 2012
5. Unravelling the Shared Genetic Mechanisms Underlying 18 Autoimmune Diseases Using a Systems Approach. Golukhadas *et al.*, *Front Immunol* **12**: 693142, 2021
6. Frequency of Polyautoimmunity in Patients with Rheumatoid Arthritis and Systemic Lupus Erythematosus, Ordoñez-Cañizares MC et al., *J Clin Rheumatol*, 2020
7. New insights into the taxonomy of autoimmune diseases based on polyautoimmunity. Rojas et al, *J Autoimmun* **126**:102780, 2022
8. Phenotype-Specific Enrichment of Mendelian Disorder Genes near GWAS Regions across 62 Complex Traits. Freund et al., *Am J Hum Genet* **103** (4): 535-552, 2018
9. Network expansion of genetic associations defines a pleiotropy map of human cell biology. Barrio-Hernandez et al., *Nat Gen* **55**: 389-398, 2023
10. Common Disease is More Complex Than Implied by the Core Gene Omnigenic Model. Wray et al., *Cell* **173** (7): 1572-1580, 2018